# Reserpine maintains photoreceptor survival in retinal ciliopathy by resolving proteostasis imbalance and ciliogenesis defects

Holly Y Chen[1‡], Manju Swaroop[2†], Samantha Papal[1†], Anupam K Mondal[1†], Hyun Beom Song[1], Laura Campello[1], Gregory J Tawa[2], Florian Regent[1], Hiroko Shimada[1§], Kunio Nagashima[3], Natalia de Val[3], Samuel G Jacobson[4#], Wei Zheng[2], Anand Swaroop[1*]

[1]Neurobiology, Neurodegeneration and Repair Laboratory, National Eye Institute, National Institutes of Health, Bethesda, United States; [2]National Therapeutics for Rare and Neglected Diseases, National Center for Advancing Translational Sciences, National Institutes of Health, Rockville, United States; [3]Electron Microscopy Laboratory, National Cancer Institute, Center for Cancer Research, Leidos Biomedical Research, Frederick National Laboratory, Frederick, United States; [4]Department of Ophthalmology, Scheie Eye Institute, Perelman School of Medicine, University of Pennsylvania, Philadelphia, United States

*For correspondence:
swaroopa@nei.nih.gov

[†]These authors contributed equally to this work

Present address: [‡]Department of Cell, Developmental and Integrative Biology, University of Alabama at Birmingham, Birmingham, United States; [§]Department of Physiology, Keio University School of Medicine, Tokyo, Japan

[#]Deceased

**Abstract** Ciliopathies manifest from sensory abnormalities to syndromic disorders with multiorgan pathologies, with retinal degeneration a highly penetrant phenotype. Photoreceptor cell death is a major cause of incurable blindness in retinal ciliopathies. To identify drug candidates to maintain photoreceptor survival, we performed an unbiased, high-throughput screening of over 6000 bioactive small molecules using retinal organoids differentiated from induced pluripotent stem cells (iPSC) of *rd16* mouse, which is a model of Leber congenital amaurosis (LCA) type 10 caused by mutations in the cilia-centrosomal gene *CEP290*. We identified five non-toxic positive hits, including the lead molecule reserpine, which maintained photoreceptor development and survival in *rd16* organoids. Reserpine also improved photoreceptors in retinal organoids derived from induced pluripotent stem cells of *LCA10* patients and in *rd16* mouse retina in vivo. Reserpine-treated patient organoids revealed modulation of signaling pathways related to cell survival/death, metabolism, and proteostasis. Further investigation uncovered dysregulation of autophagy associated with compromised primary cilium biogenesis in patient organoids and *rd16* mouse retina. Reserpine partially restored the balance between autophagy and the ubiquitin-proteasome system at least in part by increasing the cargo adaptor p62, resulting in improved primary cilium assembly. Our study identifies effective drug candidates in preclinical studies of *CEP290* retinal ciliopathies through cross-species drug discovery using iPSC-derived organoids, highlights the impact of proteostasis in the pathogenesis of ciliopathies, and provides new insights for treatments of retinal neurodegeneration.

## Editor's evaluation

This work provides an important pipeline for high-throughput screening platform to be used for drug discovery. The current data are complete. Validation of human patients-derived iPSC clones and functional assays in mice further strengthens the current conclusion.

**eLife digest** Leber congenital amaurosis (LCA) is an inherited disease that affects the eyes and causes sight loss in early childhood, which generally gets worse over time. Individuals with this condition have genetic mutations that result in the death of light-sensitive cells, known as photoreceptors, in a region called the retina at the back of the eye. Patients carrying a genetic change in the gene CEP290 account for 20-25% of all LCA.

At present, treatment options are only available for a limited number of patients with LCA. One option is to use small molecules as drugs that may target or bypass the faulty processes within the eye to help the photoreceptors survive in many different forms of LCA and other retinal diseases. However, over 90% of new drug candidates fail the first phase of clinical trials for human diseases. This in part due to the candidates having been developed using cell cultures or animal models that do not faithfully reflect how the human body works.

Recent advances in cell and developmental biology are now enabling researchers to use stem cells derived from humans to grow retina tissues in a dish in the laboratory. These tissues, known as retinal organoids, behave in a more similar way to retinas in human eyes than those of traditional animal models. However, the methods for making and maintaining human retinal organoids are time-consuming and labor-intensive, which has so far limited their use in the search for new therapies.

To address this challenge, Chen et al. developed a large-scale approach to grow retinal organoids from *rd16* mutant mice stem cells (which are a good model for LCA caused by mutations to CEP290) and used the photoreceptors from these organoids to screen over 6,000 existing drugs for their ability to promote the survival of photoreceptors. The experiments found that the drug reserpine, which was previously approved to treat high blood pressure, also helped photoreceptors to survive in the diseased organoids. Reserpine also had a similar effect in retinal organoids derived from human patients with LCA and in the *rd16* mice themselves.

Further experiments suggest that reserpine may help patients with LCA by partially restoring a process by which the body destroys and recycles old and damaged proteins in the cells. The next steps following on from this work will be to perform further tests to demonstrate that this use of reserpine is safe to enter clinical trials as a treatment for LCA and other similar eye diseases.

## Introduction

Once considered vestigial, the primary cilium has emerged as a key microtubule-based organelle that senses the external environment and modulates diverse signaling pathways in multiple tissues. Aberrant cilium biogenesis and/or ciliary transport and functions lead to numerous diseases, collectively termed ciliopathies, which manifest from sensory abnormalities to syndromic disorders with multi-organ pathologies including aberrant kidney morphogenesis, brain malformation, and congenital retinal degeneration (*Reiter and Leroux, 2017*). The mammalian retina is an extension of the central nervous system that is specialized for vision (*London et al., 2013*). The visual information is captured by rod and cone photoreceptors, integrated and processed by interneurons, and transmitted to the cortex through the retinal ganglion cells. Inability of the retinal photoreceptors to detect and/or transmit light-triggered signals is a major cause of vision impairment in retinal and macular degenerative diseases (*Veleri et al., 2015*; *Wright et al., 2010*; *Verbakel et al., 2018*). Mutations in over 200 genes can lead to inherited retinal diseases (IRDs) (RetNet, https://sph.uth.edu/retnet/), with a combined prevalence of 1/3–4000 individuals (*Verbakel et al., 2018*; *Hanany et al., 2020*). Among them, up to 20% of IRD-causing genes are involved in primary cilium biogenesis or functions (*Zelinger and Swaroop, 2018*). Due to extensive clinical and genetic heterogeneity, high inter- and intra-family variability, and incomplete penetrance (*Farrar et al., 2017*), treatment options for IRD are limited (*Garafalo et al., 2020*). Only one gene therapy drug is currently approved by FDA for Leber congenital amaurosis (LCA, MIM204000) caused by *RPE65* mutations (*Pierce and Bennett, 2015*). While the initial clinical outcomes were promising, the long-term data of this gene therapy drug are less encouraging (*Gardiner et al., 2020*). Furthermore, the development of individualized gene therapy protocols for divergent mutations in a multitude of genes for rare IRDs would be time-consuming, expensive, and labor-intensive (*Tambuyzer et al., 2020*). Thus, gene-agnostic paradigms are being developed for retinal and macular diseases using model organisms and/or stem cell-based approaches (*Scholl*

*et al., 2016*). Small-molecule or antibody drugs represent a relatively affordable and scalable option (*Tambuyzer et al., 2020*). However, over 90% of drug candidates fail in Phase I clinical trials due to the lack of effective model systems which faithfully recapitulate the pathophysiology of human diseases (*Horvath et al., 2016*). The limited number of cells in the retina and challenges in the maintenance of primary retinal cultures have also hindered the progress of therapeutic development.

LCA is a clinically severe and genetically heterogeneous group of IRDs, leads to vision loss in early childhood (*Cremers et al., 2018*). *LCA10* caused by mutations in the cilia-centrosomal gene *CEP290* (also called *NPHP6*) is one of the most common types, accounting for over 20% of patients (*den Hollander et al., 2008*). Mutations in *CEP290* exhibit pleiotropy with phenotypes ranging from LCA (affecting vision and other sensory systems) to nephronophthisis, and Joubert and Meckel syndromes involving multiple organ systems (*Sayer et al., 2006*; *Valente et al., 2006*; *McEwen et al., 2007*; *Coppieters et al., 2010*; *Drivas et al., 2015*). The large 290 kDa centrosome-cilia protein CEP290 is ubiquitously expressed and localized to the Y-links of the transition zone of primary cilia, where it acts as a hub for connecting major protein complexes and likely performs a gating function (*Sayer et al., 2006*; *Craige et al., 2010*; *Rachel et al., 2012*; *Drivas et al., 2013*). The *rd16* mouse carries an in-frame deletion in the myosin tail of CEP290, which causes a malformed connecting cilium (equivalent to the transition zone) and rudimentary outer segment (the primary cilium of photoreceptor) structure, leading to rapid degeneration of photoreceptors (*Chang et al., 2006*; *Rachel et al., 2015*). Additional studies have also demonstrated the critical role of CEP290 in cilia biogenesis (*Drivas et al., 2013*; *Tsang et al., 2008*; *Barbelanne et al., 2015*; *Wu et al., 2020*; *Prosser et al., 2022*). Thus, *LCA10* is considered a retinal ciliopathy caused by hypomorphic mutations (*den Hollander et al., 2006*; *Cideciyan et al., 2007*) that eliminate some functions of CEP290 (*Shimada et al., 2017*). Vision impairment in early childhood of *LCA10* patients imposes a considerable burden on families and society (*Leroy et al., 2021*). Notably, these patients demonstrate sparing of the foveal cones even at late stages of life, providing an attractive target for therapy (*Cideciyan et al., 2011*). Gene replacement using an AAV vector is difficult because of the large coding region of *CEP290*. Thus, other strategies including antisense oligonucleotides are currently under investigation (*Collin et al., 2012*; *Gerard et al., 2012*; *Burnight et al., 2014*; *Dooley et al., 2018*; *Dulla et al., 2018*; *Kim et al., 2018*). However, no approved treatment is currently available for alleviating vision loss due to CEP290 defects.

Generation of three-dimensional tissue organoids from induced pluripotent stem cells (iPSCs) (*Nakano et al., 2012*; *Sasai, 2013*) has revolutionized biological and disease-modeling investigations and created opportunities for high-throughput screening (HTS) to design novel treatment paradigms (*Welsbie et al., 2017*). Further refinements of human retinal organoid culture protocols have permitted high efficiency and yield, developmental staging, and higher reproducibility (*Zhong et al., 2014*; *Capowski et al., 2019*; *Kaya et al., 2019*; *Regent et al., 2022*). Biogenesis of diverse cell types in retinal organoid cultures largely recapitulates structural and temporal development of in vivo human retina and can display intrinsic light responses mimicking those of primate fovea (*Saha et al., 2022*; *Cowan et al., 2020*; *Sridhar et al., 2020*). Patient iPSC-derived human retinal organoids demonstrate disease-associated phenotypes and are being used to evaluate various therapeutic approaches (*Bell et al., 2020*; *Kruczek and Swaroop, 2020*). However, long and tedious generation protocols spanning 150+ days, inherent variability, and lack of compatible HTS assays still pose challenges to the application of human retinal organoids for developing therapies (*Struzyna and Watt, 2021*).

In this study, we aimed to establish a reliable drug discovery pipeline using an organoid-based HTS platform with the goal to identify drug candidates for maintaining photoreceptor survival in retinopathies, focusing initially on *LCA10*. For primary HTS of over 6000 bioactive small molecules, we designed survival assays using *rd16* mouse iPSC-derived retinal organoids, which showed compromised photoreceptor development and viability. An FDA-approved small molecule drug reserpine was identified as an efficacious lead compound, which also enhanced photoreceptor development/survival in *LCA10* patient iPSC-derived retinal organoids as well as in the *rd16* retina in vivo. Transcriptomic analyses of drug-treated patient organoids indicated modulation of cell survival pathways including p53 and proteostasis by reserpine. Further examination validated the mis-regulation of autophagy in patient organoids and *rd16* retina and demonstrated partial restoration of proteostasis and improved ciliogenesis after reserpine treatment. Our study thus establishes a cross-species drug discovery pipeline using organoid-based HTS and identifies a repurposing drug candidate for maintaining photoreceptor

survival in *LCA10* patients. As the action mechanisms of reserpine are partially through the modulation of proteostasis, which could be a common pathological impact of ciliopathies, reserpine, and its derivatives could potentially serve as an effective treatment for patients with other retinal ciliopathies.

## Results

### Establishment of an organoid-based HTS platform

We designed an unbiased HTS assay to identify drug candidates that might improve photoreceptor development and/or survival. For a primary screen, we chose mouse retinal organoids because of a much shorter photoreceptor differentiation time and efficient generation of rudimentary outer segments using our modified HIPRO protocol (*Chen et al., 2016a*). We decided to use a karyotypically normal iPSC line derived from the *Nrl*-GFP containing *rd16* mice (*Figure 1—figure supplement 1A*), which expresses green fluorescent protein (GFP) in all rods and carries an in-frame deletion in the myosin tail homology domain of CEP290 (*Figure 1—figure supplement 2A*). The *rd16* mice are a model of photoreceptor degeneration observed in *LCA10* patients (*Chang et al., 2006*). The iPSC lines derived from *rd16* mice displayed similar morphology, proliferation rate, and stem cell properties as those derived from the wild-type (WT) (*Figure 1—figure supplement 2B and C*). WT and *rd16* retinal organoids also showed comparable morphology at the early stages of differentiation (*Figure 1A*). However, at later stages, rod photoreceptors in *rd16* organoids displayed aberrant morphology, with malformed or missing ciliary axoneme, connecting cilium and ciliary rootlets (*Figure 1—figure supplement 2D*). We then performed flow cytometry analyses to evaluate the differentiation of GFP+ rod photoreceptors in the *rd16* organoid cultures, with the goal to identify quantifiable phenotypes for HTS. The *rd16* organoids demonstrated as much as 50% lower organoid viability and almost 60% fewer GFP+ rod photoreceptors at day (D) 32 (*Figure 1B*), the end stage of differentiation in mouse retinal organoids.

The phenotypes in GFP+ *rd16* rod photoreceptors permitted the development of an HTS screening platform to identify bioactive small-molecule candidates for augmenting rod cell differentiation and/or survival (*Figure 1—figure supplement 2E*). To avoid the impact of organoid variability on drug effects, we dissociated *rd16* organoids into single cells at D25 and performed the treatment using two-dimensional cultures from D26 to D28. These cultures recapitulated the phenotypes detected in non-dissociated organoids, and *rd16* cells displayed reduced viability and fewer GFP+ rod cells at D28 compared to the WT (*Figure 1C*).

Our drug discovery pipeline is illustrated in *Figure 1D*. In the primary screens, *rd16* retinal organoid-derived cells were treated with over 6000 small molecules; of these, 114 compounds revealed higher GFP signal intensity, indicating a positive effect on rod photoreceptor survival. We then eliminated the compounds that showed toxicity or autofluorescence even in the absence of GFP expression (false positives). Fourteen small molecule compounds were then selected for further screening to identify a lead compound.

### Selection of the lead compound

We then treated *rd16* retinal organoids with the 14 selected small molecules in a secondary assay, which started at D22 and lasted for three days to maximize the drug effect without causing toxicity (*Figure 2A*). WT, untreated, and treated *rd16* organoids were harvested at D29 to allow a sufficient period for the restoration of photoreceptors. The lead compound was selected based on the efficacy of the small molecule drugs to improve photoreceptor development and/or survival. Immunostaining of rhodopsin and S-opsin, markers of rod and cone photoreceptors respectively, was performed to quantify the drug effect (*Figure 2B*). Rhodopsin is highly expressed and polarized to the apical side of rod photoreceptors in the neural retina of WT organoids. In contrast, *rd16* photoreceptors exhibited lower expression of rhodopsin, and the polarity was considerably diminished. Two of the compounds, B05 and to some extent B03, were able to enhance rhodopsin expression, with B05 improving the polarity of expression in the neural retina as well. We note that S-cone photoreceptors are difficult to be maintained in mouse organoid culture (*DiStefano et al., 2018*); nonetheless, compounds B01 and B05 were able to augment the expression and polarity of S-opsin. Three of the 5 compounds (B01, B03, B05) that showed no toxicity revealed a dramatic increase in rod and/or cone photoreceptors in D29 *rd16* organoids (*Figure 2C*); of these, B05 was the most effective molecule and chosen as the

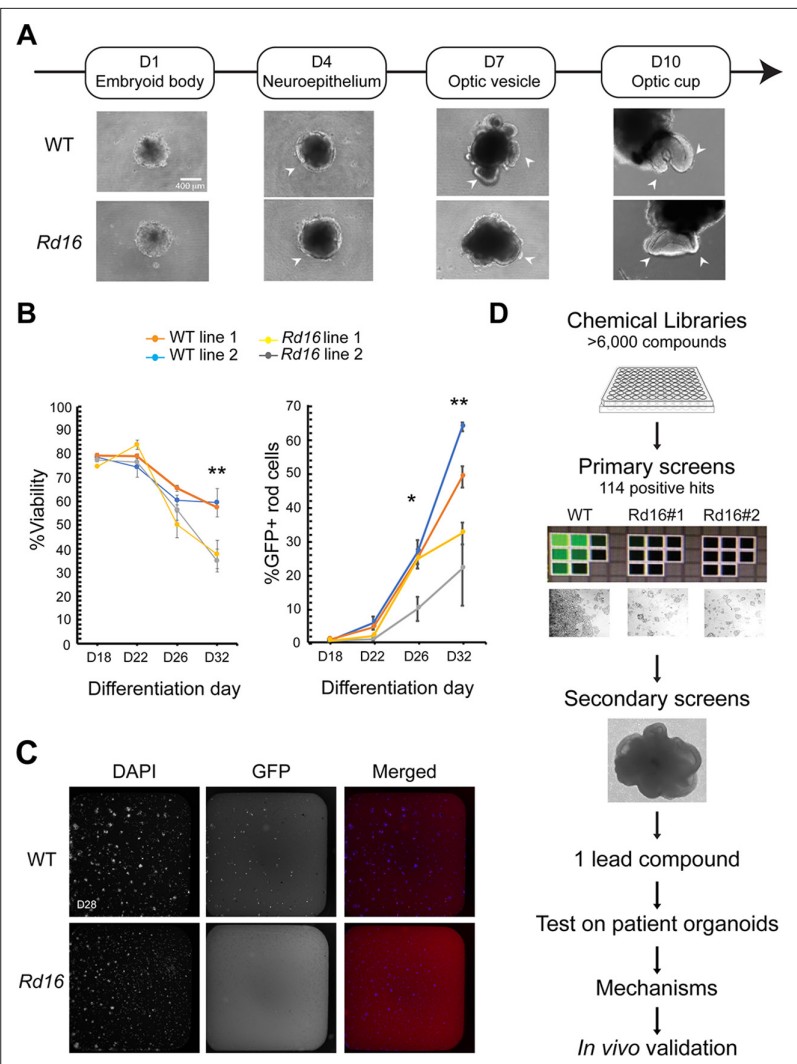

**Figure 1.** Drug discovery pipelines to identify drug candidates. (**A**) Morphology of *Nrl*-GFP wild-type (WT) and *rd16* retinal organoids differentiated from mouse-induced pluripotent stem cells (iPSC) at various differentiation time points. (**B**) Flow cytometry analysis of GFP+ rod photoreceptors and viability at different developmental stages. Each data point summarized at least three batches of independent experiments, each of which included at least 10 organoids. One-way ANOVA was performed to compare the %GFP+ cells and viability of organoids from four different cell lines. *$p<0.05$; **$p<0.01$. (**C**) Fluorescent images of dissociated day (**D**) 28 WT and *rd16* cells stained by 4',6-diamidino-2-phenylindole (DAPI) and anti-GFP antibody. (**D**) Schematic outline of the drug discovery strategy.

The online version of this article includes the following figure supplement(s) for figure 1:

**Figure supplement 1.** Karyotypes of induced pluripotent stem cell lines.

**Figure supplement 2.** Phenotypes of retinal organoids differentiated from induced pluripotent stem cells of *Nrl*-GFP *rd16* mice.

lead compound. Interestingly, the other four molecules (B01-B04) were found to be derivatives of B05, further validating our results. B05 was identified as reserpine, a small-molecule drug that has been approved by the FDA for the treatment of hypertension and schizophrenia (*Armitage et al., 1956*; *Kalliomäki and Kasanen, 1969*), thereby holding a potential for drug repurposing (*Figure 2D*).

## Validation of reserpine on *LCA10* patient organoids

To further examine the efficacy of the lead compound reserpine, we utilized iPSC-derived retinal organoids from a previously-reported *LCA10* family (*Shimada et al., 2017*), which comprised of a

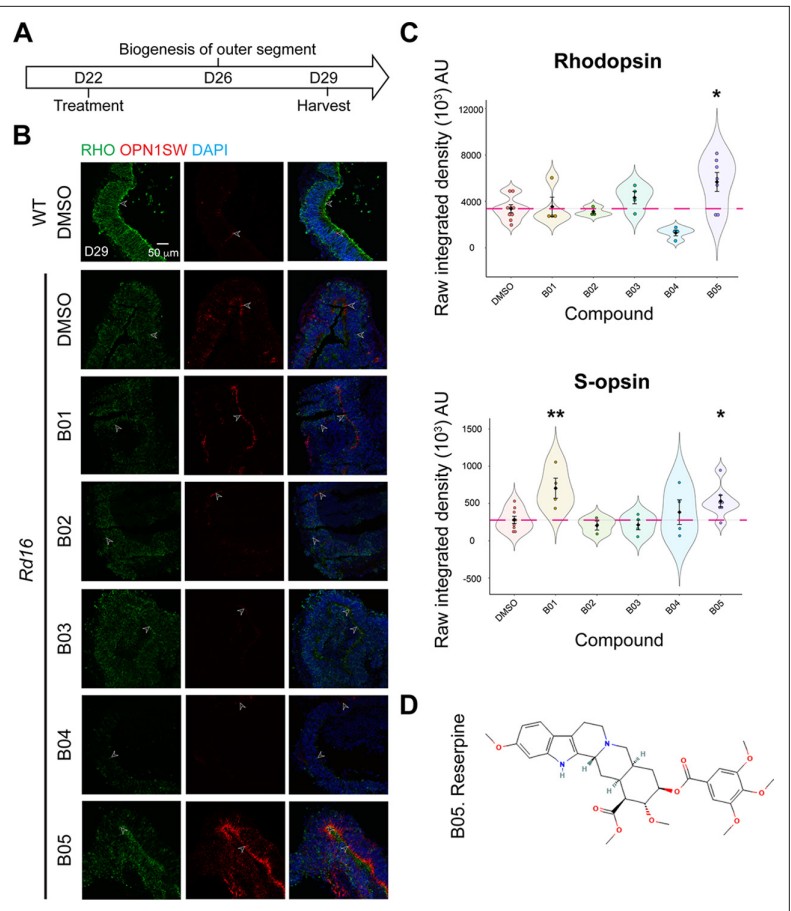

**Figure 2.** Identification of reserpine as the lead compound. (**A**) Timeline for hit validation in *rd16* retinal organoids. (**B**) Immunostaining of rod cell marker rhodopsin (RHO, green) and cone cell marker S-opsin (OPN1SW, red) in wild-type (WT) and *rd16* organoids treated with non-toxic positive hits (B01–B05). Drug vehicle dimethylsulfoxide (DMSO) was used as a control. Nuclei were stained by 4',6-diamidino-2-phenylindole (DAPI). Arrowheads indicate relevant staining. Images were representative of at least three independent experiments, each of which had at least three organoids. (**C**) Bee swarm plots show the quantification of fluorescence intensity of rhodopsin (upper) and S-opsin (lower) staining in the validation. The shape of the plot indicates the distribution of data points, which are shown by colorful circles in the center. The black diamond indicates the mean, and the error bar reveals the standard error of the mean. The pink dash line shows the mean fluorescence intensity of DMSO-treated organoids. The plot summarizes at least three independent experiments with at least three organoids in each batch. One-way ANOVA followed by the Bonferroni test was performed. *p<0.05; **p<0.01. (**D**) Chemical structure depiction of the selected lead compound reserpine.

heterozygous unaffected mother (referred to as control henceforth) and two affected compound heterozygous children (LCA-1 and LCA-2; *Figure 3—figure supplement 1A and B*). Immunoblot analysis revealed a profound reduction of full-length CEP290 in patient organoids compared to controls (*Figure 3—figure supplement 1C*). Defects in rod development and outer segment biogenesis were evident in patient organoids from D120 onwards, as revealed by the near loss of connecting cilium marker ARL13B and absence or mislocalization of rod-specific protein rhodopsin (*Figure 3—figure supplement 1D*). In concordance with the observation of retained central cones in *LCA10* patients, cone photoreceptors were only slightly compromised even at a late stage of differentiation in patient organoids.

Given that photoreceptor, outer segment biogenesis begins around D120 in organoids (*Kaya et al., 2019*) and aberrant phenotypes in patient organoids were detectable at this stage (*Figure 3—figure supplement 1D*), reserpine was added to patient organoids at D107 at a concentration of 10 µM, 20 µM, or 30 µM based on the $EC_{50}$ (half maximal effective concentration) in the primary screens (16.7 µM) (*Figure 3A*). We could detect improved connecting cilium immunostaining and

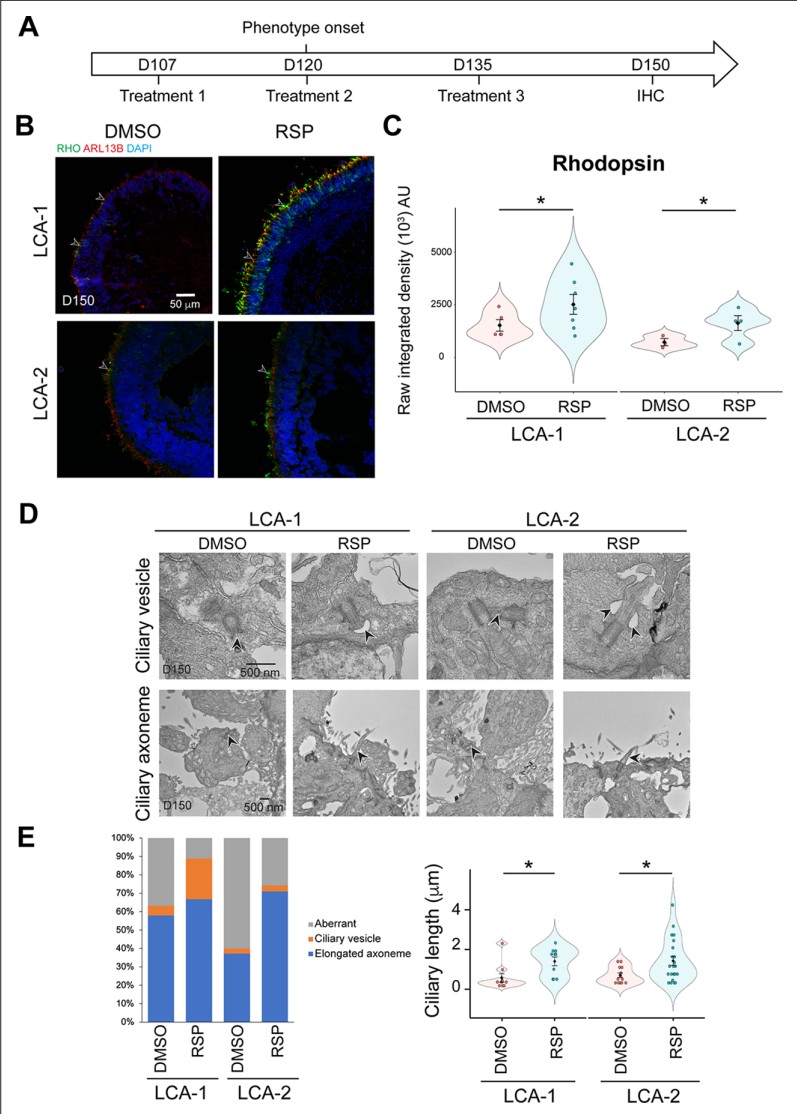

**Figure 3.** Effect of reserpine (RSP) on *LCA10* patient retinal organoids. (**A**) Timeline for RSP treatments and harvest of patient organoids. (**B**) Immunostaining of rhodopsin (RHO, green) and ARL13B (red). Nuclei were stained by 4',6-diamidino-2-phenylindole (DAPI). Arrowheads indicate relevant staining. Images were representative of at least three independent experiments, each of which had at least three organoids. (**C**) Bee swarm plots show the quantification of fluorescence intensity of rhodopsin staining. The shape of the plot indicates the distribution of data points, which are shown by colorful circles in the center. The black diamond indicates the mean, and the error bar reveals the standard error of the mean. The plot summarizes at least three independent experiments with at least one image quantified in each batch. Unpaired *t*-test was performed to compare untreated and treated groups. *p<0.05. (**D**) Transmission electron microscopy analysis of control, untreated, and treated patient organoids. Arrowheads indicate relevant staining. (**E**) Quantification of the number of the ciliary vesicles, elongated ciliary axoneme, and aberrant cilia (left) as well as the length of the primary cilia (right) in untreated and RSP-treated patient organoids. Aberrant cilia were defined as docked mother centrioles without ciliary vesicles or elongated ciliary axoneme. The data summarized at least four batches of independent experiments, each of which has at least two organoids and seven docked mother centrioles. The shape of the bee swamp plot indicates the distribution of data points, which are shown by colorful circles in the center. The black diamond indicates the mean, and the error bar reveals the standard error of the mean. Unpaired *t*-test was performed to compare untreated and treated groups. *p<0.05.

The online version of this article includes the following source data and figure supplement(s) for figure 3:

**Figure supplement 1.** Genotype and phenotype of the *CEP290*-LCA family in this study.

**Figure supplement 1—source data 1.** Overlay of the bright field and chemiluminescence images indicating the

*Figure 3 continued on next page*

*Figure 3 continued*

signal of CEP290.

**Figure supplement 1—source data 2.** Overlay of the bright field and chemiluminescence images indicating the signal of GAPDH.

**Figure supplement 2.** Effect of reserpine at various conditions in *CEP290*-LCA retinal organoids.

**Figure supplement 3.** Transmission electron microscopy analyses of control, and untreated and treated *CEP290*-LCA retinal organoids.

**Figure supplement 4.** Various cell types in reserpine (RSP)-treated *CEP290*-LCA retinal organoids.

rod development as early as D125 with the addition of 30 µM reserpine (*Figure 3—figure supplement 2A*). Even using the lowest tested dose (10 µM), we observed more polarized rhodopsin and connecting cilium at the apical side of photoreceptors in treated patient organoids, demonstrating a favorable effect of the drug (*Figure 3—figure supplement 2A*). Although variations in drug effects on organoids derived from the two patients could be observed, reserpine treatment significantly increased the fluorescence intensity of rhodopsin in retinal organoids derived from two patients (*Figure 3B, C*), suggesting an improvement of rod photoreceptor development. To avoid clonal variation, we also tested reserpine on another clone of each patient. Although statistically non-significant, these retinal organoids showed an obvious positive trend of rhodopsin staining upon reserpine treatment (*Figure 3—figure supplement 2B, C*). Such differences could be attributed to variability in response to dose and/or treatment windows of retinal organoids derived from various iPSC lines. We then performed transmission electron microscopy to uncover additional structural details of the primary cilium in photoreceptors of patient organoids after treatment. Reserpine increased the percentage of mother centrioles harboring ciliary vesicles in LCA-1 patient organoids (*Figure 3D* and *Figure 3—figure supplement 3A*), which were reportedly missing in a substantial fraction of LCA photoreceptors (*Shimada et al., 2017*). LCA-2 organoids demonstrated almost 50% more elongated ciliary axonemes after reserpine treatment though ciliary vesicles were hardly noticed. Quantification of the length of the ciliary axoneme indicated a significant increase in both patient organoids after treatment (*Figure 3E*). Notably, well-organized disc-like structures could be distinguished in treated samples (*Figure 3—figure supplement 3B*), suggesting a positive impact of reserpine treatment on photoreceptor primary cilium development.

We note that reserpine reportedly interferes with the sympathetic nervous system by inhibiting the transport of neurotransmitters into presynaptic vesicles (*Bernstein et al., 2014*); yet, even at 30 µM, no adverse effect was detected on the development of cone photoreceptors, ribbon synapses, presynaptic vesicles, bipolar cells, and Müller glia in treated patient organoids (*Figure 3—figure supplement 4*).

## Implication of cell survival and proteostasis pathways in reserpine-treated organoids

To elucidate the mechanism of action of reserpine and to gain insights into photoreceptor cell death in *LCA10*, we performed RNA-seq analyses of control and patient retinal organoids. The organoids were harvested right after the treatment with 30 µM reserpine at D150 (*Figure 4—figure supplement 1A*). Consistent with the moderate response of LCA-2 patient organoid to reserpine (*Figure 3—figure supplement 2C*), we did not identify sufficient significantly differentially expressed (DE) genes between untreated and treated groups of LCA-2 for downstream analyses (data not shown) and thereby focused only on LCA-1. Principal component analysis (PCA) revealed substantial alterations in patient organoid transcriptome compared to control and after reserpine treatment (*Figure 4A*). The largest principal component PC1 accounted for up to 36.2% of the total variation and is likely due to the drug treatment. PC2 explained another 21% of the total variation and seemed to be mainly contributed by the disease status. Notably, the reserpine-treated group was closer to the control compared to the untreated group in PC2. A total of 355 genes were significantly differentially expressed between untreated and treated organoids at thresholds of 5% FDR and twofold change (*Figure 4—source data 1*). Consistent with the PCA plot, heatmap analysis indicated that DE genes in treated patient organoids showed a relative transcriptomic shift from the untreated group toward the control ones (*Figure 4B*). However, compared to the controls, most of the downregulated genes (e.g.

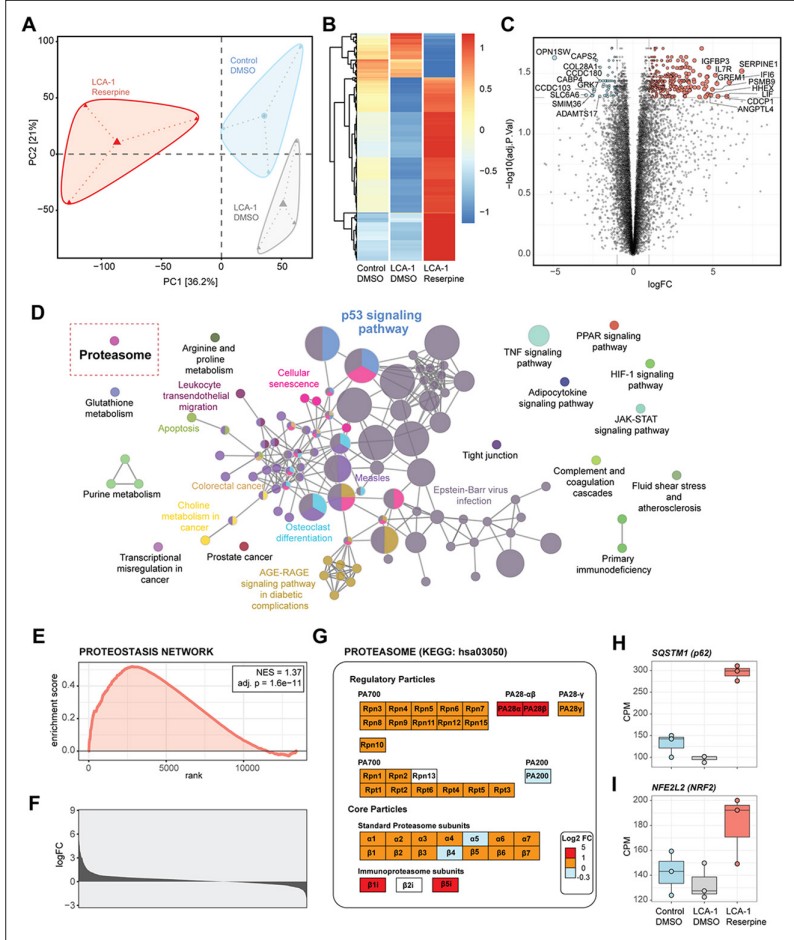

**Figure 4.** Transcriptomic upregulation of proteasomal components induced by reserpine in patient retinal organoids. (**A**) Principal component analysis (PCA) diagram of patient and control organoids shows altered retinal transcriptomes after reserpine treatment. (**B**) Drug-induced genes in patient organoids displayed specific trends compared to control organoids. (**C**) Volcano plot summarizes reserpine-induced differential gene expression changes in LCA-1 organoids. (**D**) ClueGO analysis of KEGG pathway enrichment showed overexpressed genes mapping to protein homeostasis, metabolism, and cellular signaling processes. The red rectangle highlights the 'Proteasome' pathway. (**E**) GSEA plot showing enrichment and significance of Proteostasis Network. (**F**) Histogram of the log fold change of Proteostasis Network genes upon reserpine treatment. (**G**) Proteasomal subunits responded strongly to reserpine treatment. (**H**) SQSTM1 (p62) and (**I**) NFE2L2 (NRF2), two key regulators of protein homeostasis, showed increased expression after reserpine treatment.

The online version of this article includes the following source data and figure supplement(s) for figure 4:

**Source data 1.** Differentially expressed genes in reserpine-treated retinal organoids.

**Figure supplement 1.** Reserpine treatment alters expression of retinal pathway genes.

those involved in proteostasis) in untreated LCA-1 organoids had an even higher expression upon drug treatment. Similarly, the expression of genes upregulated in untreated samples was lowered even more than the controls by reserpine treatment. Curiously, treatment of LCA-1 organoids with 10 μM or 20 μM reserpine did not yield sufficient DE genes for further analyses (data not shown).

We first analyzed the expression level of genes specific to different retinal cell types and those involved in phototransduction and photoreceptor outer segment structure/function. Reserpine treatment modulated the expression of marker genes for inner retina cell types, including Müller glia (*Figure 4—figure supplement 1B*); however, surprisingly, several rod and cone photoreceptor genes (e.g. *OPN1SW* and *GRK7*) showed lower expression (*Figure 4C*, *Figure 4—figure supplement 1C*, *Figure 4—source data 1*), with *CEP290*-associated cilia genes (*Rachel et al., 2012*) exhibiting varying changes in expression (*Figure 4—figure supplement 1D*).

As reserpine did not seem to directly regulate the expression of key components involved in disease pathology, we mapped the overexpressed genes to KEGG pathways and created an enrichment network. Using a cutoff of three genes with minimum overlap and 5% impact, the annotation network was plotted to visualize leading ontology terms (*Figure 4D*). Clusters of metabolism, proteostasis, and immune pathways, together with cell survival-related processes, were apparent in the network. To better understand functional connectivity among DE genes, we performed a Random-walktrap analysis on their protein-protein interactions network to identify co-functioning over- and under- expressing genes and identified three prominent modules (*Figure 4—figure supplement 1E*). Functional module 1 was comprised of extracellular matrix (ECM) and ECM-receptor interaction, and advanced glycation end products (AGE)-Receptor for AGE (RAGE) signaling pathway genes. Müller glia-specific genes *MMPL14*, *TIMP1*, and *VIM* were included in this module. Functional module 2 consisted of inflammation- and proteasome-related genes, which are consistent with the reported role of reserpine as an autophagy modulator (*Lee et al., 2015*). Notably, functional module 3 contained critical cell survival factors such as p53 signaling and cellular senescence-associated genes. We further investigated the trend of the p53 network to characterize its role in response to drug treatment. Expression of *TP53*, the gene coding p53, was found to increase and match the level of control organoids (*Figure 4—figure supplement 1F*). In addition, a widespread modulation of each component of the p53 signaling network was evident upon reserpine treatment (*Figure 4—figure supplement 1G*). Downstream targets of p53, including metabolic modulator TSC and mTOR complex genes, responded to reserpine and returned to the level of control organoids (*Figure 4—figure supplement 1H*). Two components of the mTORC1 signaling pathway, *RHEB*, and *LAMTOR1*, were over-expressed after the drug treatment (*Figure 4—source data 1*). As mTORC1 is a key regulator of cellular metabolism, we observed significantly elevated expression of *SLC2A1* (GLUT) which is the primary glucose transporter in neurons (*Figure 4—figure supplement 1I*).

Given the reported actions of reserpine in neuronal cells (*Lee et al., 2015*), consistent with activation of mTORC1 activation (*Kim and Guan, 2015*) and 'proteasome' subunit-encoding genes (*Figure 4D*), we performed a focused gene set enrichment analysis to test the impact on the proteostasis network (PN). A PN geneset was manually curated from KEGG and Reactome using keywords reviewed from previous publications (*Jayaraj et al., 2020*; *Klaips et al., 2018*). We detected a significant positive net enrichment of PN in patient organoids after reserpine treatment (NES = 1.37, adj. p-value = 1.6e−11; *Figure 4E*), as measured by $\log_2$ fold change in expression of all PN genes (*Figure 4F*). We also identified global overexpression of proteasomal subunits, some of which were notably induced with fold change >2 (*Figure 4G*). In concordance, expression of the key regulators of proteostasis and members of the p53 network, p62 (*SQSTM1*) and NRF2 (*NFE2L2*), were significantly augmented in reserpine-treated patient organoids (*Figure 4H, I*; *Figure 4—source data 1*).

## Restoration of proteostasis in patient photoreceptors

To experimentally examine the role of PN in the survival of *LCA10* photoreceptors, we supplemented organoid cultures with various autophagy inhibitors (MRT68921, Lys05, chloroquine, hydroxychloroquine, ROC-325) that target different steps of the autophagy pathway using the same timeline as reserpine treatment (*Figure 5—figure supplement 1A, B*). MRT68921 and Lys05 inhibit the initiation of autophagy (*Galluzzi et al., 2017*) and were highly toxic even at one-fourth of the reported $EC_{50}$. Hydroxychloroquine and a more specific and efficient small molecule autophagy inhibitor ROC-325 enhanced the number of RHO+ cells in patient organoids with more polarized localization of rhodopsin to the apical side (*Figure 5—figure supplement 1C*). Quantification of the fluorescence intensity of rhodopsin immunostaining demonstrated a consistent trend and hydroxychloroquine- and ROC-325-treated organoids had a significantly higher expression of rhodopsin (*Figure 5—figure supplement 1D*). Although statistically non-significant, chloroquine-treated organoids also showed a positive trend in rhodopsin expression (*Figure 5—figure supplement 1D*), providing further evidence in support of the autophagy pathway as a target for designing therapies.

To further determine the role of autophagy, we evaluated the level of cargo receptor sequestosome 1 (SQTM1) or p62, which recognizes cellular components and helps in the formation of autophagosomes for proteolysis (*Mizushima, 2007*; *Figure 5A*). We noted that LCA-1 patient organoids had a significantly lower level of p62 compared to the control, and reserpine treatment significantly elevated p62 levels by more than twofold (*Figure 5B*), consistent with previous studies (*Lee et al.,*

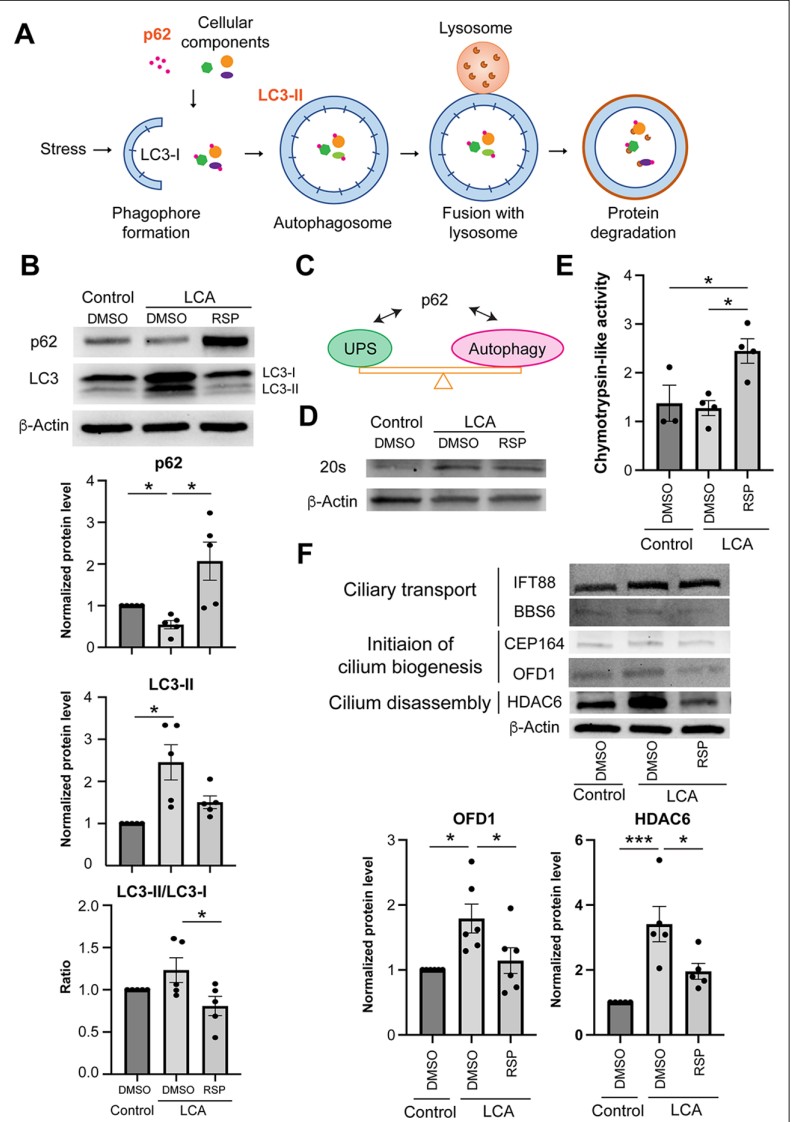

**Figure 5.** Proteostasis Network in patient organoid cultures in response to reserpine treatment. (**A**) Schematic diagram of autophagy.(**B**) Immunoblot analyses and quantification of autophagy cargo adaptor p62 and autophagosome marker LC-II in control and patient organoids treated with reserpine (RSP). (**C**) Schematic diagram showing the proteome balance between ubiquitin-proteasome system (UPS) and autophagy is mediated through p62 as documented in the literature. (**D**) Immunoblot analysis of the 20S proteasome in control, DMSO-, and RSP-treated cultures. β-Actin was used as the loading control. (**E**) Proteasomal chymotrypsin-like activity in organoids. (**F**) Immunoblot analyses and quantification of key regulators of cilium assembly/disassembly in control, untreated, and RSP-treated patient organoids. The drug vehicle dimethylsulfoxide (DMSO) was added to the cultures in the untreated group at the same volume as the drugs. β-Actin was used as the loading control. The histograms summarize data in at least three batches of experiments, each of which had at least three retinal organoids per group. Each dot in the histogram shows data in one batch of experiment and are presented as mean ± standard deviation. One-way ANOVA followed by Tukey's test. *p<0.05; ***p<0.005.

The online version of this article includes the following source data and figure supplement(s) for figure 5:

**Source data 1.** Overlay of the bright field and chemiluminescence images indicating the signal of p62.

**Source data 2.** Overlay of the bright field and chemiluminescence images indicating the signal of LC3.

**Source data 3.** Overlay of the bright field and chemiluminescence images indicating the signal of β-Actin.

**Source data 4.** Overlay of the bright field and chemiluminescence images indicating the signal of 20S proteosome and β-Actin.

*Figure 5 continued on next page*

*Figure 5 continued*

**Source data 5.** Overlay of the bright field and chemiluminescence images indicating the signal of IFT88.

**Source data 6.** Overlay of the bright field and chemiluminescence images indicating the signal of BBS6.

**Source data 7.** Overlay of the bright field and chemiluminescence images indicating the signal of CEP164.

**Source data 8.** Overlay of the bright field and chemiluminescence images indicating the signal of OFD1.

**Source data 9.** Overlay of the bright field and chemiluminescence images indicating the signal of HDAC6.

**Source data 10.** Overlay of the bright field and chemiluminescence images indicating the signal of β-Actin.

**Figure supplement 1.** Effect of autophagy inhibitors on patient organoids.

**Figure supplement 2.** Bafilomycin A1 (BafA1) and Tubastatin A treatment on patient organoids.

**Figure supplement 2—source data 1.** Overlay of the bright field and chemiluminescence images indicating the signal of LC3.

**Figure supplement 2—source data 2.** Overlay of the bright field and chemiluminescence images indicating the signal of p62.

**Figure supplement 2—source data 3.** Overlay of the bright field and chemiluminescence images indicating the signal of β-Actin.

**Figure supplement 3.** LCA2-derived retinal organoids in response to reserpine (RSP) treatment.

**Figure supplement 3—source data 1.** Overlay of the bright field and chemiluminescence images indicating the signal of p62.

**Figure supplement 3—source data 2.** The size of the protein ladders, p62, and relevant sample identity are labeled.

**Figure supplement 3—source data 3.** Overlay of the bright field and chemiluminescence images indicating the signal of OFD1.

**Figure supplement 3—source data 4.** Overlay of the bright field and chemiluminescence images indicating the signal of HDAC6.

*2015*; *Bresciani et al., 2018*). Significantly higher levels of LC3-II in patient organoids also indicated accumulation of autophagosomes (defects in autophagy), and reserpine treatment revealed a trend of decreasing LC3-II levels (though not statistically significant) in patient organoids (*Figure 5B*). The comparable ratio of LC3-II/LC3-I indicated a similar rate of autophagosome formation between control and patient organoids, suggesting that increased LC3-II levels in patient organoids could be due to overexpression of autophagy pathway components. Consistent with the reported function of reserpine as an autophagy inhibitor, the treated patient organoids had a significantly lower LC3-II/LC3-I ratio, although we did not detect a significant difference between the control and patient organoids (*Figure 5B*).

We then applied Bafilomycin A1 (BafA1), an inhibitor of autophagosome-lysosome fusion, to patient organoid cultures using the same timeline as reserpine treatment (*Figure 5—figure supplement 2A*). A short 6-hr treatment with BafA1 did not alter the autophagy pathway in the control, as shown by comparable levels of p62 and LC3-II as well as LC3-II/LC3-I ratio (*Figure 5—figure supplement 2B*); however, patient organoids demonstrated a significant (up to 70%) increase of LC3-II and LC3-II/LC3-I ratio, suggesting an elevated autophagic flux in patient organoids, consistent with the rescue by reserpine treatment.

The ubiquitin-proteasome system (UPS) and autophagy are the two key pathways in proteostasis and are reported to interact through p62 (*Kumar et al., 2022*; *Liu et al., 2016*; *Figure 5C*). We, therefore, investigated the response of UPS in patient organoids upon reserpine treatment. Untreated patient organoids revealed a high level of 20S proteasome, which could be barely detected in the control (*Figure 5D*). Nonetheless, both the control and untreated patient organoids showed comparable total catalytic activity and reserpine significantly elevated the proteasome activity (*Figure 5E*). These results suggest that an increase in 20S expression is likely a compensatory mechanism for compromised proteasome activity in patient organoids and that reserpine facilitates the clearance of accumulated cellular components and/or autophagosomes (*Figure 5B*).

We were intrigued by the reported links of autophagy to primary cilium biogenesis (*Pampliega et al., 2013*; *Tang et al., 2013*; *Yamamoto and Mizushima, 2021*) and, therefore, looked at the expression of key regulators involved in ciliary transport, cilium assembly, and disassembly (*Figure 5F*).

We identified higher expression of OFD1 (oral-facial-digital syndrome 1), which when eliminated by autophagy is shown to promote ciliogenesis (*Tang et al., 2013*), in patient organoids compared to the control even though autophagy activity was higher in the latter (*Figure 5F*). This apparent ambiguity could be due to variations in autophagic adapter machinery for cargo identification. Nonetheless, reserpine treatment reduced OFD1 levels in patient organoids and should facilitate the initiation of cilia biogenesis. Another key regulator, histone deacetylase 6 (HDAC6), which deacetylates micro-tubules and destabilizes the primary cilium for disassembly (*Simões-Pires et al., 2013*; *Pugacheva et al., 2007*), was significantly elevated in patient organoids compared to the control. Dramatic reduction of HDAC6 by reserpine (*Figure 5F*) would also have a favorable impact on cilia biogen-esis. Consistent with this hypothesis, the addition of a selective HDAC6 inhibitor Tubastatin A to patient organoids improved rod photoreceptor development as shown by a higher number of RHO + cells (*Figure 5—figure supplement 2C*). Tubastatin A treatment enhanced the polarity of not only rhodopsin but also S-opsin, likely due to increased stability of intracellular microtubules and improved intracellular trafficking. Quantification of the fluorescence intensity of rhodopsin indicated a significant increase of rhodopsin expression in tubastatin A-treated organoids (*Figure 5—figure supplement 2D*), suggesting a favorable effect on rod photoreceptor development probably through the inhibi-tion of HDAC6.

To validate the effect of reserpine, we assessed the autophagy machineries and relevant ciliary markers in a less responsive LCA-2 clone F upon reserpine treatment. Consistent with LCA-1, reser-pine treatment significantly increased the p62 level but did not alter LC3-II/LC3-I ratio in LCA-2 organoids (*Figure 5—figure supplement 3A, B*). Although statistically non-significant, the level of autophagosome showed a decreasing trend in organoids treated with reserpine (*Figure 5—figure supplement 3A, B*). As retinal organoids contain abundant retinal cell types besides photoreceptors, we performed immunostaining of p62 on control, untreated, and treated LCA1 and LCA2 organoids. As shown in *Figure 5—figure supplement 3C*, p62 staining could barely be observed in both patient organoids compared to the control. Reserpine treatment dramatically increased p62 staining in both photoreceptors and other retinal cell types. We also quantified the level of OFD1 and HDAC6, the two ciliary regulators that mediated the favorable effect of reserpine on outer segment biogenesis (*Figure 5—figure supplement 3D*). We observed a significant decrease of HDAC6 in LCA2 organ-oids upon reserpine treatment. Although statistically non-significant, a lower expression of OFD1 was shown in reserpine-treated LCA2 organoids. Therefore, the action mechanisms of reserpine are independent of the responsiveness of patient organoids.

## Maintenance of photoreceptor survival in *rd16* mice in vivo

We then performed intravitreal injection of 40 µM reserpine into *rd16* mouse eyes at postnatal day (P)4, and the retinas were harvested at P21 (*Figure 6A*). Reserpine was able to maintain photore-ceptor survival, with a significantly thicker outer nuclear layer (ONL) in both the central and peripheral retina (*Figure 6B*). As a small molecule drug, reserpine is highly diffusive, and we observed the effect of the drug in the contralateral eyes of the treated mice (*Figure 6—figure supplement 1*). Indeed, bilateral therapeutic effects have been reported following unilateral injection in clinical treatments (*Michalska-Małecka et al., 2016*; *Rouvas et al., 2009*; *Calvo et al., 2016*; *Zlotcavitch et al., 2015*; *Rotsos et al., 2014*). Reserpine-treated *rd16* mice showed a trend of increase in scotopic a-wave and a significant increase in scotopic b-wave (*Figure 6C and D*), suggesting functional recovery of rod photoreceptors upon reserpine treatment. No statistically significant difference was found in photopic a-wave or b-wave (*Figure 6D*). Notably, no systemic toxicity was observed in the treated mice. Injec-tion of reserpine into WT mice did not lead to structural or functional alterations (*Figure 6—figure supplement 2A and B*), suggesting reserpine to be a safe treatment option.

Consistent with the effects observed on human retinal organoids, reserpine treatment increased p62 levels (*Figure 7A and B*), dramatically reduced the 20S proteasome (*Figure 7C*), and significantly enhanced the proteasome activity (*Figure 7D*). Further analyses confirmed an improvement in the structure of photoreceptor outer segments of the treated *rd16* retina. Reserpine partially restored outer segment axonemes that were largely missing in untreated mouse retina (*Figure 7E*). In addition, substantial ciliary rootlets were conspicuous in the inner segment (shown by GFP) and extended into the ONL. Phototransduction proteins including rhodopsin and Pde6β were transported to the outer segment (*Figure 6—figure supplement 3A*). In addition, the treatment with 40 µM reserpine did not

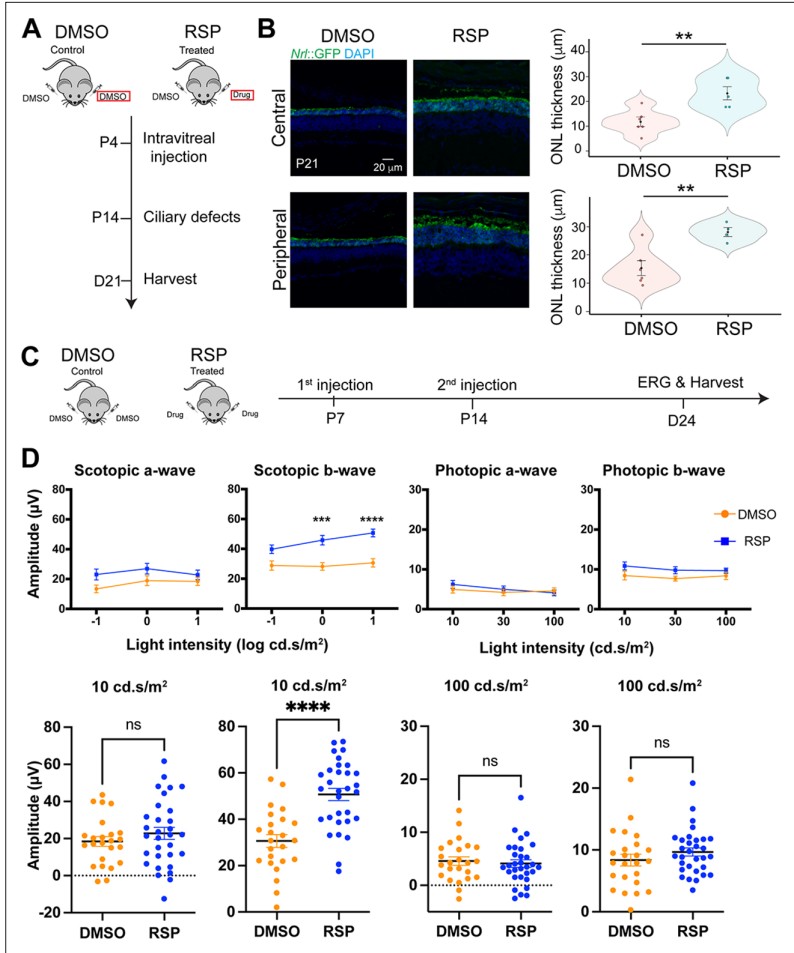

**Figure 6.** Neuroprotective effect of reserpine (RSP) in *rd16* mouse retina. (**A**) Timeline of in vivo intravitreal injection for quantification of outer nuclear layer (ONL), morphology, and immunohistochemistry studies. The eyes used for subsequent analyses were highlighted by red rectangles. (**B**) Immunostaining of dimethylsulfoxide (DMSO)- and reserpine (RSP)-treated *rd16* retina (left) and quantification of the GFP+ outer nuclear layer (ONL) thickness (right). The shape of the bee swamp plot indicates the distribution of data points from four different litters, in which at least one mouse was assessed. Data points are shown in colorful circles in the center. The black diamond indicates the mean, and the error bar reveals the standard error of the mean. Unpaired *t*-test was used to compare the mean. *p<0.05. (**C**) Timeline of in vivo intravitreal injection for electroretinography (ERG) studies. (**D**) Scotopic and photopic electroretinogram responses in DMSO- and RSP-treated *rd16* mice. The lower panel shows individual values of a- and b-wave amplitudes at 10 cd.s/m² (scotopic) or at 100 cd.s/m² (photopic). ERG was measured in both eyes of at least one mouse from five different litters in each group (12 DMSO-treated and 16 RSP-treated mice). Data were expressed as mean ± SEM, and the Mann-Whitney U test was used to compare DMSO- and RSP-treated groups. ***p<0.001; ****p<0.0001.

The online version of this article includes the following source data and figure supplement(s) for figure 6:

**Source data 1.** Individual values of a- and b-wave amplitudes (µV) of scotopic and photopic electroretinogram responses in DMSO- and RSP-treated *rd16* mice.

**Figure supplement 1.** Evaluation of reserpine (RSP) injection on photoreceptor layer in *rd16* mouse retina.

**Figure supplement 2.** Evaluation of reserpine (RSP) injection on photoreceptor structure and function in the wild-type (WT) mouse retina.

**Figure supplement 2—source data 1.** Individual values of a- and b-wave amplitudes (µV) of scotopic and photopic electroretinogram responses in DMSO- and RSP-treated WT mice.

**Figure supplement 3.** Effect of reserpine (RSP)-injection on distinct cell types in *rd16* mouse retina.

**Figure supplement 3—source data 1.** Overlay of the bright field and chemiluminescence images indicating the signal of Gfap.

*Figure 6 continued on next page*

*Figure 6 continued*

**Figure supplement 3—source data 2.** Overlay of the bright field and chemiluminescence images indicating the signal of g-Tubulin.

alter the morphology of other retinal cell types or structures including bipolar neurons, amacrine cells, and Müller glia (*Figure 6—figure supplement 3B and C*). Notably, the retinal stress marker Gfap was significantly increased in untreated *rd16* retina as compared to the WT and was significantly reduced upon reserpine treatment (*Figure 6—figure supplement 3D and E*), suggesting a favorable effect of reserpine to retinal homeostasis.

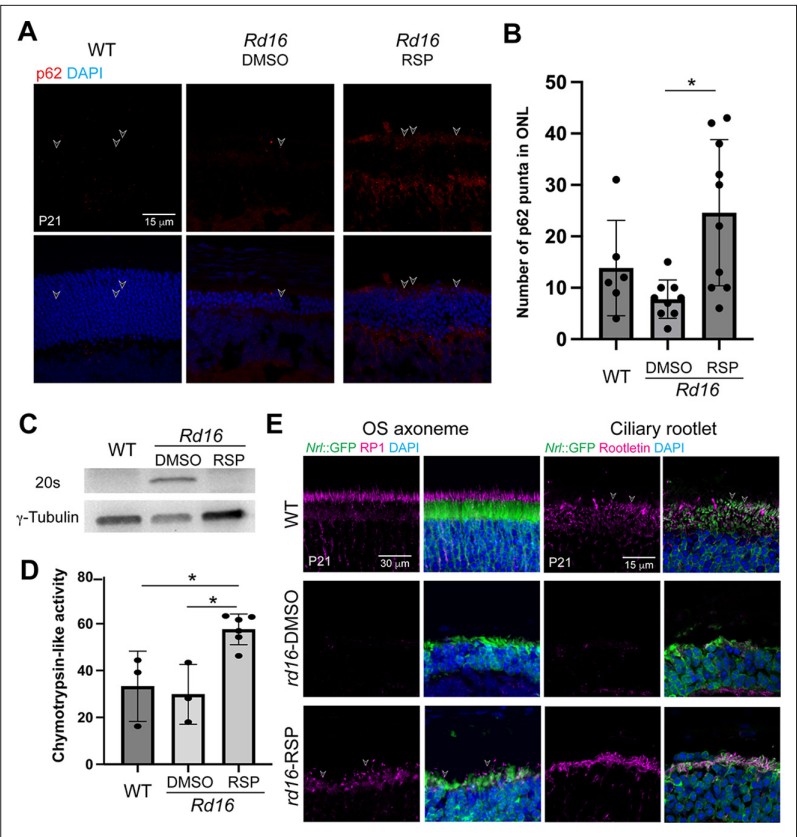

**Figure 7.** *Rd16* mouse retina in response to reserpine (RSP) treatment. (**A**) p62 level in dimethylsulfoxide (DMSO)- and RSP-treated photoreceptors shown by immunostaining. (**B**) Quantification of p62 puncta. At least three non-overlap regions were quantified in at least one mouse from at least two different litters. One-way ANOVA followed by Tukey's test was used to compare different groups. *p<0.05. (**C**) Western blot analysis of 20S proteasome in wild-type (WT), DMSO-, and RSP-treated retina of *rd16* mice. γ-Tubulin was used as the loading control. (**D**) Proteasomal chymotrypsin-like activity in *rd16* retina. Data in the histogram were summarized from at least three mice from different litters and are presented as mean ± standard deviation. One-way ANOVA followed by Tukey's test was used to compare different groups. *p<0.05. (**E**) Immunostaining of outer segment (OS) axoneme marker RP1 (magenta, left panel) and ciliary rootlet marker Rootletin (magenta, right panel). In (**A**) and (**E**), nuclei were stained by 4',6-diamidino-2-phenylindole (DAPI). Arrowheads indicate relevant staining. Images were representative of at least three mice from different litters.

The online version of this article includes the following source data for figure 7:

**Source data 1.** Overlay of the bright field and chemiluminescence images indicating the signal of 20S proteosome.

**Source data 2.** Overlay of the bright field and chemiluminescence images indicating the signal of γ-Tubulin.

# Discussion

Repurposing of existing drugs for new unrelated clinical modalities provides an excellent opportunity for alleviating patient sufferings in a timely and cost-effective manner (*Begley et al., 2021*). HTS of approved small molecule drugs can be based on simple assays that target the modulation of a disease-related phenotype or molecule. The use of patient-derived iPSCs and their derivatives, including differentiated cell types in two-dimensional or organoids in three-dimensional cultures, have substantially enhanced the prospects of successful drug discovery (*Struzyna and Watt, 2021*; *Engle and Puppala, 2013*). Exciting screening platforms are now being applied to retinal cells and organoids for drug discovery (*Welsbie et al., 2017*; *Vergara et al., 2017*). We note the success of cross-species screening, in which the drug candidates identified using one species could be effective in another species (*Zhang et al., 2021a*). Given the 'orphan' status of IRDs, extensive genetic and phenotypic heterogeneity, and predominantly early dysfunction/death of rods, we designed a simple assay using GFP-tagged rods from iPSCs of a mouse mutant that phenocopies *LCA10* and established an HTS platform to identify small-molecule drugs to maintain photoreceptor survival. As a two-dimensional primary or stem cell-derived cultures demonstrate a relatively lower variation and have generated promising candidates in screenings (*Welsbie et al., 2017*; *Kost-Alimova et al., 2020*; *Mills et al., 2019*; *Chen et al., 2016b*), we dissociated mouse retinal organoids into single cells and performed HTS in two-dimensional cultures. The identified lead compound reserpine was subsequently verified to be effective on patient organoids and in mouse mutant retina in vivo, suggesting the feasibility of our approach for drug discovery.

To our knowledge, there is no other study testing reserpine in other retinal diseases. Reserpine was approved for the treatment of hypertension in 1955 and later for the treatment of schizophrenia (https://www.pubchem.gov/); however, many better-tolerated and more potent hypertensive medications have become available during the last few decades. Though potential side effects of inhibition of presynaptic vesicle formation and consequently depression have been described for reserpine treatment (*Yohn et al., 2017*), we did not observe adverse consequences on ribbon synapses or presynaptic vesicles in treated organoids or retinal function in WT mouse retina in vivo (*Figure 3—figure supplement 4* and *Figure 6—figure supplement 2B*). Our results are consistent with a previous study demonstrating the little effect of reserpine on central sympathetic activity in cats (*Iggo and Vogt, 1960*). Additionally, the doses of reserpine for the treatment of hypertension or schizophrenia in adults range from 8 mg/kg to 80 mg/kg for a 60 kg adult (*Cheung and Parmar, 2023*), which is higher than the dose we used in mouse studies (9.6 mg/kg). Reserpine has also been intramuscular or intravenous injection at a dose of 1 mg/kg to assess the modulation of the outcome of intraocular pressure drug and allergic inflammation of the eye in rabbits (*Okada and Shimada, 1980*; *Delamere and Williams, 1985*). No adverse effects have been reported in both studies. Besides, local delivery by intravitreal injection or via eye drops should be sufficient for reserpine to elicit its effects in the retina due to the highly diffusive properties of small-molecule drugs. Thus, we suggest that reserpine could be a safe therapeutic approach for the treatment of *LCA10* and probably other retinal ciliopathies. Future studies will focus on toxicity evaluation as well as on identifying more potent and less toxic derivatives of reserpine to initial clinical trials.

Transcriptomic analyses have permitted us to interrogate potential mechanism(s) of reserpine action in patient organoids and implicated signaling pathways involved in immune response (e.g. primary immunodeficiency, complement, and coagulation cascade), cell survival, and cell death (e.g. p53 signaling pathway, cellular senescence, apoptosis), metabolism (e.g. glutathione metabolism, purine metabolism), and proteostasis. Indeed, these pathways have highly intricate relationships. The p53 protein acts as a sensor of stress conditions and can act both as a transcriptional activator or repressor to promote cell death and cell survival decisions (*Kruiswijk et al., 2015*; *Liu et al., 2021*; *Wylie et al., 2022*). Gene profiles of treated patient organoids revealed a significant increase of *TP53* as well as its downstream targets including key cellular metabolic regulator TSC and mTOR complexes. More importantly, the expression level of genes involved in TSC and mTOR complexes returned to the level of control organoids, suggesting a positive impact of reserpine in restoring photoreceptor metabolism. As both TSC knockout and mTOR complex 1 activation are reported to maintain photoreceptor survival (*Venkatesh et al., 2015*; *Venkatesh et al., 2016*), the trends in our experimental system suggest a cell survival effect of the p53 pathway through the modulation of cell metabolism. In addition, the expression of several Müller glia-specific genes was altered by reserpine,

with GFAP levels reduced in mouse retina in vivo (*Figure 6—figure supplement 3D and E*). Müller glia is believed to play a major role in reactive gliosis and likely adapt their transcriptome to support photoreceptors in retinal degeneration (*Tomita et al., 2021*; *Palko et al., 2022*). Yet, we are unsure whether reserpine directly acts on Müller glia to augment photoreceptor survival, or its response is a consequence of improved microenvironment and reduced retinal stress. Notably, the primary cilium appears to have a role in Müller glia maturity and functions in primary cultures (*Ferraro et al., 2015*). Whether and how Müller glia are affected in *LCA10* or other degenerative diseases and their role in photoreceptor survival are active areas of investigation.

CEP290 is localized at the connecting cilium of photoreceptors and serves as a protein complex hub to control the biogenesis and function of outer segments. Hypomorphic CEP290 in the mouse degenerative model *rd16* leads to malformed connecting cilia, compromised development of outer segments, and mislocalization of phototransduction machineries (*Chang et al., 2006*; *Rachel et al., 2015*). Comparable phenotypes are observed in *LCA10* patient-derived retinal organoids (*Figure 3—figure supplement 1D*; *Shimada et al., 2017*; *Parfitt et al., 2016*). Ciliary defects lead to the accumulation of outer segment proteins (e.g. opsin) in the endoplasmic reticulum (ER) and subsequently induce the unfolded protein response (*Reiter and Leroux, 2017*), which plays a crucial role in the regulation of autophagy, a key quality control mechanism for degradation and recycling of components in response to starvation, growth factor deprivation, ER stress, and pathogen infection (*He and Klionsky, 2009*). Although autophagy is a protective surveillance mechanism, chronic protein stress, such as in the case of IRD and aging retina, may dysregulate the degradation machineries and subsequently lead to cell death due to disruption of cellular proteostasis (*Weinberg et al., 2022*; *Qiu et al., 2019*). Loss of proteostasis has been shown to attribute to the pathogenesis of retinal and macular degeneration including IRD and age-related macular degeneration (AMD) (*Weinberg et al., 2022*; *Faber and Roepman, 2019*; *Tolone et al., 2022*). Cellular proteostasis is largely maintained by the autophagy-lysosome pathway and the ubiquitin-proteasome system (UPS). Multiple studies reveal that modulation of autophagy or UPS could be a promising therapeutic approach to maintain photoreceptor survival in IRD (*Qiu et al., 2019*; *Sen et al., 2021*; *Yao et al., 2018*) and AMD (*Zhang et al., 2021b*; *Vessey et al., 2022*). A favorable effect of autophagy inhibition has also been reported for retinal ganglion cell survival (*Zaninello et al., 2020*). Interestingly, both proteasome inhibition and activation have been shown to be able to maintain photoreceptor survival and function in a mouse degenerative model P23H carrying misfolding of rhodopsin (*Qiu et al., 2019*; *Sen et al., 2021*). This suggests a more complicated regulatory mechanism in the restoration of proteostasis. We observed increased autophagic flux in *LCA10* patient organoids together with reduced p62 and accumulation of autophagosome (*Figure 5B* and *Figure 5—figure supplement 2B*). Though it seems counterintuitive to reduce autophagic flux as the accumulated protein would aggravate disease pathology, we note that reserpine treatment facilitated the clearance of autophagosome at least partially through the activation of the UPS in both retinal organoids and in vivo mouse retina (*Figure 5 B-E* and *Figure 7 A-D*). Although no significant difference in 20 S proteasome activity was found in patient-derived organoids or *rd16* mouse retina as compared to the control/WT (*Figures 5E and 7D*), a higher expression of 20 S proteasome was observed (*Figures 5D and 7C*), suggesting a compromised proteosome activities in patient-derived organoids and *rd16* retina. The UPS and autophagy reciprocally regulate the activity of each other through the common cargo adaptor p62 (*Kumar et al., 2022*; *Liu et al., 2016*). Consistent with a previous study (*Lee et al., 2015*), reserpine increased the p62 level, which should facilitate the activation of UPS. Consistently, we observed clearance of accumulated autophagosome even though an autophagy inhibitor reserpine was applied (*Figure 5B*). Therefore, we propose that, instead of modulating a single pathway, the proteostasis network should be considered as a whole system in developing therapies (*Figure 8*). Further studies are needed to dissect the proteostasis network and understand the mechanisms of recognition of targeted components for degradation.

Improved outer segment biogenesis is another prominent effect of reserpine in our study, which is consistent with a previous study showing reserpine induces rod outer segment elongation (*Kralj and Pipan, 1998*). As a key signaling regulator, the primary cilium is essential for the activation of starvation-induced autophagy through modulation of the Hedgehog pathway and other components involved in autophagy, which can eliminate components in intraflagellar transport (*Pampliega et al., 2013*) as well as remove OFD1 from distal appendages to initiate cilium biogenesis (*Tang et al., 2013*). Thus, the outcome of the crosstalk between ciliogenesis and autophagy seems to largely depend on cell

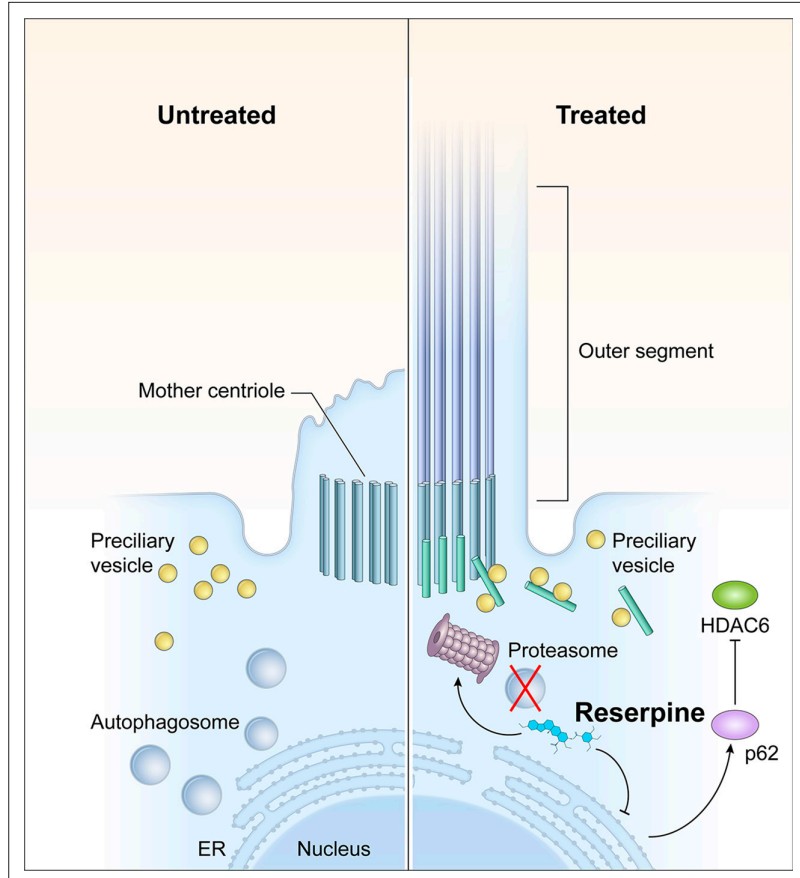

**Figure 8.** Action mechanisms of reserpine in photoreceptor survival. *CEP29*0 mutations lead to defects in the outer segment biogenesis and consequently, ciliary vesicles carrying building blocks of the primary cilium and ciliary proteins are accumulated in patient photoreceptors, leading to the activation of autophagy to degrade unwanted materials. As an autophagy inhibitor, reserpine downregulates autophagy and increases the p62 level in photoreceptors. As p62 is a mediator and cargo adaptor of the ubiquitin-proteasome system and autophagy, upregulation of p62 not only activates the 20 S proteasome to facilitate the clearance of the accumulated autophagosome but also facilitates the degradation of histone deacetylase 6 (HDAC6), which deacetylates microtubules. Removal of HDAC6 in photoreceptors thus should improve the stability of intracellular microtubules and facilitate the transport of pre-ciliary vesicles to the mother centriole for outer segment biogenesis.

type and physiological context. Our observation of high OFD1 levels in patient organoids compared to the control, despite an elevated autophagic flux, suggests alternative mechanisms. HDAC6 has also been demonstrated to promote autophagosome maturation (*Lee et al., 2010*; *Chang et al., 2021*) as well as deacetylate microtubules to impede intracellular trafficking, which is a key driver of primary cilium disassembly. Higher expression of HDAC6 in patient organoids could explain both the failure of docking of pre-ciliary vesicles (*Shimada et al., 2017*) and elevated autophagic flux in patient photoreceptors. HDAC6 is shown to be a target of p62 for degradation, and an increase in p62 in photoreceptors can reduce HDAC6 leading to improved outer segment length (*Toulis et al., 2020*). We suggest that reserpine-induced p62 levels can facilitate the degradation of HDAC6 and augment the intracellular transport of preciliary vesicles to the mother centriole for cilia biogenesis (*Figure 8*).

In conclusion, we identified a repurposing small-molecule drug reserpine to maintain photoreceptor survival in retinal ciliopathies, specifically *LCA10*, and at least partially act by restoration of proteostasis in photoreceptors. Reserpine has been evaluated in a human context (using patient organoids) and mouse retina in vivo and thus holds promise in future clinical studies. As the loss of proteomic homeostasis is a major cause of multiple retinal degenerative diseases (*Faber and Roepman, 2019*), reserpine and its derivatives have clinical potential for gene-agnostic therapies. Despite the identification of pathways associated with reserpine action, it is hard to dissect the function of each pathway as

they are highly intertwined with each other. We realize that transcriptomic analysis to elucidate drug effect was performed with organoids of a single patient (LCA-1). In addition, the drug candidate we identified using mouse organoids is effective on patient organoids and mouse retina in vivo; however, certain pathogenetic pathways specific to humans might be missing in the drug target search by the current HTS strategy. A major future direction will be to evaluate the effect and safety of different dosages of reserpine on photoreceptor survival and function in multiple degenerative mouse models and using patient iPSC-derived retinal organoids of other ciliopathies in IRD.

## Materials and methods

### Animals

*B6.Cg-Cep290rd16/Boc* mice (Strain #: 012283; RRID: IMSR_JAX:012283) were obtained from the Jackson Laboratory and crossed to *Nrlp-eGFP* mice (*Akimoto et al., 2006*) to generate *Nrlp-EGFP*; *Cep290*$^{rd16/rd16}$ mice (referred to as *rd16*). The absence of the *rd8* mutation in the colony was assessed by PCR. All animal procedures were approved by the Animal Care and Use committee of the National Eye Institutes (Animal study protocol NEI-650) and adhered to ARVO Statement for the Use of Animals in Ophthalmic and Vision Research. Mice were housed in an atmosphere controlled-environment (temperature: 22°C ± 2°C, humidity: 30–70%), under a 12 hr dark/12 hr light cycle, and supplied with food and water ad libitum. Food, water, and nesting material were changed weekly.

### Mouse and human pluripotent cell lines

The mouse *Nrl*-GFP WT and *rd16* iPSC lines were obtained by infection of the E14.5 *Nrlp-eGFP* and *rd16* mouse embryonic fibroblasts with Dox-inducible lentiviral vectors carrying *Pou5f1*, *Sox2*, *Klf4*, and *Myc* genes individually, as previously described (*Chen et al., 2016a*; *Mookherjee et al., 2018*). To obtain human iPSC lines, fibroblasts of the familial control, two clones of *LCA10* patient 1 (annotated as LCA-1 and LCA-1-clone C), and two clones of *LCA10* patient 2 (annotated as LCA-2 and LCA-2-clone D) were isolated from skin biopsies and reprogrammed using integration-free Sendai virus by the iPSC Core Facility at National Heart, Lung, and Blood Institute of the National Institutes of Health using the established protocol (*Shimada et al., 2017*; *Beers et al., 2015*). All mouse and human iPSC lines were karyotypically normal except the *rd16* line 2, which had one missing sex chromosome and thus, was not used for subsequent high-throughput screening and secondary validation assays (*Figure 1—figure supplement 1*). All cultures were tested for mycoplasma contamination by real-time PCR periodically.

### Maintenance of iPSC

The WT and *rd16* iPSC lines were maintained as previously described (*Chen et al., 2020*). Briefly, the iPSC lines were maintained on feeder cells (Millipore) in a maintenance medium constituted by Knockout DMEM (ThermoFisher Scientific), 1 x MEM non-essential amino acids (NEAA) (Sigma), 1 x GlutaMAX (ThermoFisher Scientific), 1 x Penicillin-Streptomycin (PS) (ThermoFisher Scientific), 2000 U/ml leukemia inhibitory factor (LIF) (Millipore), and 15% ES cell-qualified fetal bovine serum (FBS) (ThermoFisher Scientific) at 37 °C, 5% $CO_2$. Full media change was performed every day, with 55 µM β-Mercaptoethanol (2-ME) (ThermoFisher Scientific) freshly added. Cells were passaged using TrypLE Express (ThermoFisher Scientific) every two days.

Human iPSC lines were maintained in Essential 8 (E8) (ThermoFisher Scientific) on growth factor-reduced (GFR) Matrigel (Corning)-coated plates, with media fully changed daily. Cells were maintained at 37 °C, 5% $O_2$, and 5% $CO_2$ and passaged at 60–80% confluency using the EDTA-based dissociation method (*Kelley et al., 2020*).

### Differentiation of mouse and human retinal organoids

The modified HIPRO protocol was used to differentiate mouse iPSCs into retinal organoids (*Chen et al., 2016a*; *Chen et al., 2020*). At differentiation day (D)0, iPSCs were plated in low adhesion U-shaped 96-well plate (Wako) at a density of 3000–5000 cells per well in 100 µl retinal differentiation medium consisting of GMEM (ThermoFisher Scientific), 1 x NEAA, 1 x sodium pyruvate (Sigma) and 1.5%(v/v) knockout serum replacement (KSR) (ThermoFisher Scientific). At D1, 240 µl GFR-Matrigel (>9.5 mg/ml) was diluted in 1.8 ml retinal differentiation medium and 20 µl diluent was aliquoted to

each well of the 96-well plate. Retinal organoids from one 96-well plate were transferred to a 100 mm poly(2-hydroxyethyl methacrylate) (Sigma)-coated petri dish with 12 ml DMEM/F12 with GlutaMAX, 1x N2 supplement, and 1 x PS. The media were replaced by DMEM/F12 with GlutaMAX, 1 x PS, 1x N2 supplement, 1 mM taurine (Sigma), 500 nM 9-*cis* retinal (Sigma), and 100 ng/ml insulin-like growth factor 1 (IGF1) (ThermoFisher Scientific) at D10, and half-media change was performed every other day, with 55 µM 2-ME freshly added to the media. From D26 onward, 1 x NEAA, 1 x B27 supplement without Vitamin A (ThermoFisher Scientific), and 2%(v/v) FBS (ThermoFisher Scientific) were added to the culture. Half-media exchanges were performed every two days, with 55 µM 2-ME freshly added to the media. The cultures were incubated in 5% $O_2$ from D0 to D10 and in 20% $O_2$ from D10 onwards.

Human retinal organoid differentiation was performed as previously described (*Regent et al., 2020*). Briefly, small clumps dissociated from iPSCs in one well of a six-well plate were resuspended in E8 medium supplemented with 10 µM Y-27632 (Tocris) and transferred into one 100 mm polyHEMA-coated petri dish for embryoid body (EB) formation. Media were supplied with neural induction media (NIM) (DMEM/F-12 (1:1)) (ThermoFisher Scientific), 1x N2 supplement, 1 x NEAA, 2 µg/ml heparin (Sigma) at D1 and D2 at a ratio of 3:1 and 1:1, respectively, and fully switched to NIM at D3. D7 EBs from one 100 mm petri dish were plated onto one GFR Matrigel-coated 60 mm dish and cultured in NIM, with media changed every 2–3 days. In the application requiring a large-scale production of retinal organoids, nicotinamide was added to the culture to reach a final concentration of 5 mM from D0 to D8 (*Regent et al., 2022*). NIM was replaced by a 3:1 retinal induction medium (RIM) consisting of DMEM/F-12 supplemented with 1 x B27 without Vitamin A (ThermoFisher Scientific), 1% antibiotic-antimycotic solution (ThermoFisher Scientific), 1% GlutaMAX and 1 X NEAA at D16 and media change was performed every day until D28, on which the adherent cells were scraped off into small clumps (<5 mm²) and split into two polyHEMA-coated 100 mm Petri dishes. The floating clumps were cultured in RIM supplemented with IGF1 (ThermoFisher Scientific) and 1 mM taurine (Sigma), with 55 µM 2-ME freshly added. From D38 onward, 10% fetal bovine serum was added to RIM supplemented with 20 ng/ml IGF-1, and 1 mM taurine. 1 µM 9-*cis* retinal was freshly supplemented to the cultures during media change from D63 to D91. From D91 till the end of differentiation, the concentration of 9-*cis* retinal was reduced to 0.5 µM and B27 without Vitamin A was replaced by N2. Once scraped, media were half changed every 2–3 days, with IGF1, taurine and 9-*cis* retinal freshly added to the media under dim light environment.

## Dissociation of mouse retinal organoids into single cells

Mouse retinal organoids were transferred into a 15 ml centrifuge tube using wide-bored transfer pipets and washed one time with 10 ml 1 x PBS (ThermoFisher Scientific). Prewarmed 1 ml 0.25% trypsin-EDTA was then added, and the tube was incubated at 37 °C for 10 min, with pipetting up and down for 10 times using P1000 pipetman at 5- and 10 min incubation. 10 ml retinal maturation media (DMEM/F12 with GlutaMAX, 1 x PS, 1x N2 supplement, 1 mM taurine, 500 nM 9-*cis* retinal, 1 x NEAA, 1 x B27 supplement without Vitamin A, 2%(v/v) FBS) was added to the tube. The tube was inverted several times before centrifugation at 200 g for 5 min. After removal of the supernatant, the cell pellet was resuspended in 1 ml retinal maturation media and filtered through a 40 µm cell strainer (BD Bioscience) before proceeding to subsequent analyses.

## Compound libraries

We used three different small molecule libraries for our screening efforts. The library of 1280 pharmacologically active compounds (LOPAC 1280) consists of a collection of small molecules with characterized biological activities commonly used to test and validate screening assays. The library of U.S. Food and Drug Administration (FDA)–2800 approved drugs was set up internally. The compounds were first dissolved in 100% DMSO to generate a stock concentration with a final concentration of 10 mM and subsequently diluted into seven concentrations and dispensed into 1536-well plates. The NCATS Mechanism Interrogation Plate (MIPE) 5.0 library contains a collection of 1912 compounds (approved for clinical trials).

## HTS assays

To optimize the assay for HTS, D25 WT, and *rd16* mouse retinal organoids were dissociated into single-cell suspension and plated in 5 µl retinal maturation medium II at a concentration of 3000, 4000,

and 5000 cells per well in a 1536-well plate. After culturing in a 37 °C humidified incubator overnight, 5000 cells/well yielded an optimal plating density for both WT and *rd16* cultures and were selected for subsequent assays.

In the screening experiments for drug candidates, dissociated D25 WT-positive control and *rd16* mouse retinal organoids were filtered through a 35 µm cell strainer and dispensed into a 1536-well plate at a density of 5000 cells/well in 5 µl freshly prepared retinal maturation medium II using the Multidrop Combi Dispenser (Thermo Scientific). Cells were incubated in a 37 °C humidified incubator overnight for cell recovery and attachment to the plates. Compounds in the 1536-well drug source plates were added to the 1536-well assay plates at a volume of 23 µl/well using an NX-TR pintool station (WAKO Scientific Solutions). Cells were treated with compounds for 48 hr, followed by the addition of 0.5 µl quencher, and GFP and DAPI signal intensities were quantified using the acumen Cellista (TTP, Labtech). The positive hits with a significant increase of GFP compared to the DMSO-treated cells were retested using dissociated retinal organoids from iPSCs without a GFP marker to remove autofluorescent false positives. The remaining drug candidates were further tested using a full 11-concentration setting (1:3 serial dilutions starting at 10 mM) in triplicate plates to prioritize the hits based on efficacy.

The primary and confirmatory screening data were analyzed using software developed internally at the NIH Chemical Genomics Center (NCGC) (*Wang et al., 2010*). The plate data was processed as follows: We first ran quality control on the plate data by visual inspection, masking wells showing erroneous signals, e.g., localized groupings of wells exhibiting enhanced or inhibited signals. We then performed intra-plate normalization as follows:

$$Efficacy = 100 \times \frac{GFP(RD16+C) - GFP(RD16+DMSO)}{GFP(WT+DMSO) - GFP(RD16+DMSO)} \tag{1}$$

$$Toxicity = -100 \times \frac{DAPI(RD16+C) - DAPI(RD16+DMSO)}{DAPI(RD16+DMSO)} \tag{2}$$

$$Fluorescence = 100 \times \frac{GFP(Parental+C) - GFP(Parental+DMSO)}{GFP(WT+DMSO) - GFP(Parental+DMSO)} \tag{3}$$

Where:

WT = D25 WT retinal organoid cells with GFP marker = GFP+
RD16=D25 *rd16* retinal organoid cells with GFP marker = GFP−
Parental = Retinal organoids from iPSCs without GFP marker = GFP−
C = compound tested
GFP = GFP (channel 488) signal intensity
DAPI = DAPI (channel 405) signal intensity

When the data is normalized in this fashion, efficacy, toxicity, and fluorescence range from 0 to 100%. With this normalization completed, we compute dose-response curves as follows:

$$A(C_i) = A_0 + \frac{(A_\infty - A_0)}{[1+(10^{(Log(Ac50)-Log(c_i))})^n]} \tag{4}$$

Where $C_i$ = the *i'th* concentration, $A(C_i)$ = activity (efficacy, toxicity, or fluorescence) at concentration *i*, $A_0$ = activity at zero concentration, $A_\infty$ = activity at infinite concentration, and $EC_{50}$ = the concentration giving a response half-way between the fitted top (100%) and bottom (0%) of the curve.

Compounds are desired that have high efficacy ($EC_{50efficacy}$ = small, $A_{\infty, efficacy}$ = high), low toxicity, ($EC_{50toxicity}$ = large), and low fluorescence ($EC_{50fluorescence}$ = large).

$$EC_{50toxicity}/EC_{50efficacy} \gg 1 \tag{5}$$

$$EC_{50flurorescence}/EC_{50efficacy} \gg 1 \tag{6}$$

$$A_{\infty,efficacy} = high \tag{7}$$

In addition, the dose-response curves for each measured quantity must be well behaved, i.e., (i) curves exhibit values near zero at low concentration, (ii) values increase with increasing concentration,

(ii) curves show a well-defined inflection point at the $EC_{50}$ concentration, and (iv) curves show a well-defined plateau at high concentration (**Wang et al., 2010**).

## Immunoblot analysis

At least three organoids in each batch were homogenized in 100 µl of RIPA buffer (Sigma) supplemented with 1 x protease inhibitor (Roche) and 1 x phosphatase inhibitor (Roche). The lysate was agitated at 4 °C for half an hour, before centrifugation at 12,000 g for 10 min at 4 °C. The supernatant was either stored at –80 °C until use or quantified by Pierce bicinchoninic acid (BCA) protein assay (ThermoFisher Scientific). Approximately 20 µg protein was diluted 4:1 in reducing 4 x Laemmli buffer (Biorad) and boiled for 10 min. The samples were separated at 150–180 V for 1 hr on 4–15% precast polyacrylamide gel (Biorad) and transferred to polyvinylidene fluoride (PVDF) membranes using a TransBlot Turbo Transfer System (Biorad). After blocking in 5% milk or 5% bovine serum albumin (BSA) for 1 hr at room temperature, the blots were incubated in antibody cocktails (Key Resources Table) overnight in 1% milk or BSA in 1 X TBS-T at 4 °C overnight with gentle agitation. Membranes were subsequently washed in 1 X TBS-T for three times, 10 min each, and incubated in 1 X TBS-T with horseradish peroxidase-conjugated secondary antibodies (1:5000) for 1 hr at room temperature, followed by another three 10 min wash. Before imaging, the membranes were exposed to Super-Signal West Pico enhanced chemiluminescence (ECL) solution (ThermoFisher Scientific) for 5 min, and chemiluminescence was captured using a Bio-Rad ChemiDoc touch (Bio-Rad).

## Chymotrypsin-like proteasome activity assay

The chymotrypsin-like protease activity associated with the proteasome complex in organoids or mouse retina was quantified using the Proteasome 20 S Activity Assay Kit (Sigma) following the manufacturer's protocol. In short, at least three retinal organoids or one mouse retina were homogenized in 55 µl PBS mixed with 55 µl reconstituted Proteasome Assay Loading Solution and incubated on ice for 30 min. After centrifuging at 10,000 g for 10 min at 4 °C, 10 µl supernatant was taken for BCA assay to determine the protein amount and the remaining 100 µl was transferred to one well of black/clear 96-well plate for incubation at 37 °C. Protease activity of individual samples was measured by the fluorescence intensity ($\lambda$ ex = 480–500 nm/ $\lambda$ em = 520–530 nm) normalized to the protein amount.

## Transmission electron microscopy

Retinal organoids were processed for transmission electron microscopy (TEM) analysis as previously described (**Shimada et al., 2017**). Briefly, the organoids were fixed in 4% formaldehyde and 2% glutaraldehyde in 0.1 M cacodylate buffer, pH 7.4 (Tousimis) for 2 hr at room temperature, followed by three washes in cacodylate buffer before further fixation in osmium tetroxide (1% v/v in 0.1 M cacodylate buffer; Electron Microscopy Sciences) for 1 hr at room temperature. The organoids were then washed in the same buffer for three times, followed by one wash in acetate buffer (0.1 M, pH 4.2), and *en-bloc* staining in uranyl acetate (0.5% w/v; Electron Microscopy Sciences) in acetate buffer for 1 hr at room temperature. The samples were dehydrated in a series of ethanol solutions (35%, 50%, 75%, 95%, and 100%) and then by propylene oxide. The samples were subsequently infiltrated in a mixture of propylene oxide and epoxy resin (1:1, v/v) overnight, embedded in a flat mold with pure epoxy resin, and cured at 55 °C for 48 hr. 70–80 nm sections were made with an ultramicrotome (UC 7) and diamond knife (Diatome), attached on a 200-mesh copper grid, and counter-stained in the aqueous solution of uranyl acetate (0.5% w/v) and then lead citrate solutions. The thin sections were stabilized by carbon evaporation before the EM examination. The digital images were taken using a digital camera equipped with an electron microscope (H7650) (AMT).

## Flow cytometry

After dissociating retinal organoids, the cells were resuspended in DPBS (ThermoFisher Scientific) containing 1 mM EDTA (Millipore) and filtered through a 40 µm cell strainer. 4',6-diamidino-2-phenylindole (DAPI) (ThermoFisher Scientific) was added to the samples before being analyzed by FACSAriaII (BD Bioscience). Cell viability was evaluated by integrity (gated by DAPI), size (gated by forward scatter, FSC), and granularity (gated by side scatter, SSC). WT retinal organoids without a GFP marker on the same differentiation day were used to set the gating for GFP + cells.

## RNA extraction and library preparation

Total RNA was purified from homogenized retinal organoids using TriPure isolation reagent (Roche). Quality of isolated RNAs was assessed using Bioanalyzer RNA 6000 nano assays (Agilent) and high-quality total RNA (RNA integrity number ≥7.5) was used for the construction of the mRNA sequencing library. 100 ng total RNA was used to construct the strand-specific libraries using TruSeq RNA Sample Prep Kit-v2 (Illumina).

## RNA-seq data analysis

RNA sequencing was performed as described (**Kim et al., 2016**). Paired-end 125 bp reads were generated using Illumina sequencing. Reads were quality checked and mapped to the reference transcriptome using Kallisto. Alignments were imported into R using tximport for downstream analyses. The edgeR and limma pipeline were employed for differential expression analysis, while pathway annotation was performed using ClueGO (**Bindea et al., 2009**). Protein interaction data were obtained from STRING (v11.5) with evidence cut off of 700, via the R package STRINGdb (**Szklarczyk et al., 2021**). Network analysis was performed using the *igraph* package [https://igraph.org/] and genes were mapped to pathways using gProfileR (**Raudvere et al., 2019**). Proteostasis network genes were manually collected by pooling KEGG and Reactome pathways with keywords from **Jayaraj et al., 2020** and (**Klaips et al., 2018**), and the proteosome map was adapted from KEGG (hsa03050). Gene set enrichment analysis was done using the R package fgsea. All other plotting and analyses, unless otherwise mentioned, were performed using tidyverse and base R packages.

## Immunohistochemistry

Mouse and human organoids were fixed in 4% PFA (Electron Microscopy Sciences) for 1 hr at room temperature, washed one time, and cryoprotected in 15% sucrose for at least 2 hr at room temperature, followed by 30% sucrose at 4 °C overnight. The next day, organoids were embedded in Shandon M-1 Embedding Matrix (ThermoFisher Scientific). The blocks were sectioned at 10 µm thickness and incubated at room temperature for at least 1 hr, before immunostaining or storage at −80 °C. After incubating in blocking solution (5% donkey serum in PBS) for 1 hr at room temperature, the sections were supplied with diluted primary antibodies (Key Resources Table) at 4 °C overnight. After three 10 min washes in PBS, Species-specific secondary antibodies conjugated with Alexa Fluor 488, 568, or 637, together with DAPI, were diluted in blocking solution (1:500; ThermoFisher Scientific) and incubated with the sections for 1 hr at room temperature. After washing in PBS for three times, 10 min each, the sections were mounted for imaging.

For mouse retinal sections, *rd16* mice of both sexes were injected at postnatal (P) day 4 and recovered at P21 after euthanasia using a $CO_2$ atmosphere. Eyes were enucleated before being pierced in the center of the cornea using a 26-gauge needle. Eyecups were then incubated for 15 min in 4% paraformaldehyde (PFA). The cornea and lens were then dissected, and the eyecups were placed in 4% PFA for 15 min at room temperature (RT) before proceeding to cryoprotection in 20 and 30% sucrose-PBS at 4 °C for 1 hr and overnight, respectively. Eyecups were then quickly frozen in Shandon M-1 Embedding Matrix (Thermo Fisher Scientific) and cut at 12 µm. Retinal sections were washed twice in PBS and then blocked in PBS containing 5% Donkey serum and 0.3% Triton X-100 (PBST) for 1 hr at room temperature. Slides were incubated overnight at 4 °C with the primary antibody diluted in PBST at an appropriate concentration (see Key Resources Table). Sections were then washed three times with PBS and incubated with a secondary antibody and 1 µg/ml 4,6-diamidino-2-phenylindole (DAPI) for 1 hr at room temperature. After three washes in PBS, the sections were mounted in Fluoromount-G (SouthernBiotech).

## Image acquisition and analysis

Bright field images were taken using an EVOS XL Core Cell Imaging System (ThermoFisher). Fluorescence images were acquired with LSM-880 confocal microscope (Zeiss) with Zen software. FIJI and Photoshop CC 2019 software was used for image export, analyzing, and processing. Rhodopsin and S-opsin fluorescent intensity of organoid sections were quantified with FIJI using the maximum intensity projections of z-stack images. Multi-channel RGB (red, green, blue) images were separated into 8-bit grayscale images, and regions of interest were identified by applying the "Moment" threshold algorithm of FIJI. The same threshold algorithm was used for all images. Area, raw, and integrated

fluorescence intensity in each image were then quantified with Fiji and plotted using RStudio version 1.1.463.

## Intravitreal injection

Animal experiments were conducted in the animal facility at the National Eye Institute, the National Institutes of Health. The facility approved the animal care and experimental procedures used in this study (NEI-ASP650).

For evaluating the effect of reserpine, we performed two sets of experiments and examined mouse retinal function by electroretinography (ERG) and structure by histology and/or immunohistochemistry. In the first set, the mice were injected with reserpine at P4 and harvested at P21. In the other set, the mice were given two injections of the drug at P7 and P14, and the ERG was recorded at P24 before harvesting the retina for structural analysis. For intravitreal injections at P4, pups of either sex of *Nrl*-GFP *rd16* mice were anesthetized on ice and their eyelids were opened using a 30-gauge needle to gently expose the eye. Ketoprofen (0.5 µl of 1 mg/ml in PBS) was administered for analgesia. DMSO or reserpine (0.5 µl) was blindly injected using a glass micropipette (World Precision Instruments) produced by a Flaming-Brown micropipette puller (Sutter Instruments) and connected to an Eppendorf Femtojet air compressor (Eppendorf). For intravitreal drug injections at P7 and P14, mouse pups were anesthetized by intraperitoneal injection of ketamine (50 mg/kg body weight) and xylazine (5 mg/kg body weight). DMSO or reserpine (0.5–1 µl) was intravitreally injected using a Hamilton syringe with a 34-guage needle. Mice with eye/lens damage following the injection were excluded from further analyses.

## ERG recordings

ERG responses were recorded as previously published (*Hargrove-Grimes et al., 2020*). Briefly, after an overnight dark adaptation, mice were anesthetized with an intraperitoneal injection of ketamine (100 mg/kg) and xylazine (10 mg/kg). Eyes were dilated by topical administration of tropicamide (1% wt/vol, Alcon) and phenylephrine (2.5% wt/vol, Alcon), and the body temperature was maintained at 37 °C by a heating pad. After placing the reference electrode in the mouth and applying hypromellose ophthalmic demulcent solution (2.5% wt/vol, Gonak; Akorn) to each eye, gold wire loop electrodes were placed on the center of each cornea and ERG responses were recorded with an Espion E2 Visual Electrophysiology System (Diagnosys). Scotopic responses were recorded at increasing light intensities from 0.0001 to 10 cd·s/m$^2$ with inter-stimulus intervals ranging from 5 s to 60 s depending on stimulus intensity. After a light adaptation for 2 min, photopic responses were recorded at increasing light intensities from 0.3 to 100 cd·s/m$^2$ under a background light saturating rod function. The a-wave amplitudes were measured from the baseline to the negative trough and the b-wave amplitudes were measured from the a-wave trough to the wave peak. Mice with eye/lens damage following the injection were excluded from further analyses.

## Statistics

Based on the initial data, the sample size of animals used in each group was determined by an unpaired *t*-test to be less than six mice in each group (http://www.biomath.info/). For experiments involvng mouse or human retinal organoids, at least four independent experiments, each of which had at least three organoids, were performed unless specified. For animal studies, at least two retinas from one mouse of at least four different litters were evaluated. All data were expressed as mean ± standard deviation (SD) unless specified. An unpaired *t*-test was used to compare the mean between the two groups. For the comparison of three or more groups, one-way AVOVA was performed. Results with a *p*-value <0.05 were considered statistically significant.

## Material availability statement

All materials including the pluripotent stem cell lines and animals are available upon request. A Material Transfer Agreement complying with the guidelines of the National Institutes of Health has to be established before shipment.

# Acknowledgements

We thank Tiansen Li, Linn Gieser, Matthew Brooks, Ryan A Kelley and Trupti Shetty (National Eye Institute), and Wenwei Huang (National Center for Advancing Translational Sciences) for technical support and insightful discussions. We are grateful to Rafael Villasmil, Julie Laux, Jessica Albrecht, and Jacqueline Minehart (Flow Cytometry Core Facility of the National Eye Institute), Lijin Dong, Pinghu Liu, and Jingqi Lei (Genetic Engineering Core Facility of the National Eye Institute), Jizhong Zou and Jeanette Beers (iPSC Core Facility at National Heart, Lung, and Blood Institute), Sandra Burkett (Molecular Cytogenetic Core Facility at National Cancer Institute), and Alan Hoofring and Ethan Tyler (Division of Medical Arts at National Institutes of Health) for assistance in various aspects of research. These studies were supported by the National Eye Institute Intramural Research Program (ZIAEY000450 and ZIAEY000546) and National Center for Advancing Translational Sciences Intramural Research Program (ZIATR000018-06) and utilized the high-performance computational capabilities of the Biowulf Linux cluster at the National Institutes of Health. We wish to dedicate this manuscript to the memory of Dr. Samuel G Jacobson, a dedicated clinician scientist, who unfortunately passed away during the preparation of the revised manuscript.

## Additional information

### Competing interests

Holly Y Chen, Manju Swaroop, Samantha Papal, Anupam K Mondal, Gregory J Tawa, Wei Zheng, Anand Swaroop: Listed as inventor on a patent application related to the small molecules in this study by National Institutes of Health (PCT/US2021/040157). The other authors declare that no competing interests exist.

### Funding

| Funder | Grant reference number | Author |
| --- | --- | --- |
| National Eye Institute | Z01EY000546 | Anand Swaroop |
| National Eye Institute | Z01EY000450 | Anand Swaroop |
| National Center for Advancing Translational Sciences | ZIATR000018-06 | Wei Zheng |

The funders had no role in study design, data collection and interpretation, or the decision to submit the work for publication.

### Author contributions

Holly Y Chen, Conceptualization, Formal analysis, Validation, Investigation, Methodology, Writing – original draft, Writing – review and editing; Manju Swaroop, Conceptualization, Resources, Investigation, Methodology, Writing – review and editing; Samantha Papal, Investigation, Visualization, Methodology, Writing – review and editing; Anupam K Mondal, Data curation, Software, Formal analysis, Writing – review and editing; Hyun Beom Song, Formal analysis, Investigation, Methodology, Writing – original draft; Laura Campello, Formal analysis, Investigation, Methodology, Writing – review and editing; Gregory J Tawa, Formal analysis, Writing – review and editing; Florian Regent, Methodology, Writing – review and editing; Hiroko Shimada, Natalia de Val, Samuel G Jacobson, Resources, Writing – review and editing; Kunio Nagashima, Resources, Investigation, Methodology, Writing – review and editing; Wei Zheng, Resources, Supervision, Funding acquisition, Writing – review and editing; Anand Swaroop, Conceptualization, Resources, Supervision, Funding acquisition, Writing – original draft, Project administration, Writing – review and editing

### Author ORCIDs

Samantha Papal http://orcid.org/0000-0001-9417-6215
Hyun Beom Song http://orcid.org/0000-0002-3500-2984
Florian Regent http://orcid.org/0009-0003-2716-1584
Wei Zheng http://orcid.org/0000-0003-1034-0757
Anand Swaroop http://orcid.org/0000-0002-1975-1141

## Ethics

All animal procedures were approved by the Animal Care and Use committee of the National Eye Institutes (Animal study protocol NEI-650) and adhered to ARVO Statement for the Use of Animals in Ophthalmic and Vision Research.

## Decision letter and Author response

Decision letter https://doi.org/10.7554/eLife.83205.sa1
Author response https://doi.org/10.7554/eLife.83205.sa2

---

# Additional files

## Supplementary files
• MDAR checklist

## Data availability

All data needed to evaluate the conclusions in the paper are present in the paper and/or the Supplementary Materials. RNA-seq data are available through GEO accession #206959.

The following dataset was generated:

| Author(s) | Year | Dataset title | Dataset URL | Database and Identifier |
|---|---|---|---|---|
| Chen HY, Swaroop M, Papal S, Mondal AK, Tawa G, Regent F, Shimada H, Nagashima K, de Val N, Jacobson SG, Zhang W, Swaroop A | 2023 | Photoreceptor survival in CEP290-retinopathy by Reserpine involves modulation of proteostasis | http://www.ncbi.nlm.nih.gov/geo/query/acc.cgi?acc=GSE206959 | NCBI Gene Expression Omnibus, GSE206959 |

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

# Appendix 1

## Key Resource Table

**Appendix 1—key resources table**

| Reagent type (species) or resource | Designation | Source or reference | Identifiers | Additional information |
|---|---|---|---|---|
| Cell line (*M. musculus*) | *Nrl*-GFP WT | PMID: 27667917 | | Reprogrammed from mouse embryonic fibroblasts by lentivirus carrying Yamanaka factors |
| Cell line (*M. musculus*) | *Nrl*-GFP rd16 line 1 | PMID: 30332642 | | Reprogrammed from mouse embryonic fibroblasts by lentivirus carrying Yamanaka factors |
| Cell line (*M. musculus*) | *Nrl*-GFP rd16 line 2 | PMID: 30332642 | | Reprogrammed from mouse embryonic fibroblasts by lentivirus carrying Yamanaka factors |
| Cell line (*Homo-sapiens*) | Control | PMID: 28700940 | | Reprogrammed from control fibroblasts by Sendai virus carrying Yamanaka factors |
| Cell line (*Homo-sapiens*) | LCA-1 | PMID: 28700940 | | Reprogrammed from control fibroblasts by Sendai virus carrying Yamanaka factors |
| Cell line (*Homo-sapiens*) | LCA-2 | In this paper | | Reprogrammed from control fibroblasts by Sendai virus carrying Yamanaka factors and maintained in Swaroop lab in National Eye Institute, National Institutes of Health, United States |
| Cell line (*Homo-sapiens*) | LCA-1-clone C | In this paper | | Reprogrammed from control fibroblasts by Sendai virus carrying Yamanaka factors and maintained in Swaroop lab in National Eye Institute, National Institutes of Health, United States |
| Cell line (*Homo-sapiens*) | LCA-2-clone F | PMID: 28700940 | | |
| Antibody | anti-RHO (mouse monoclonal) | Gift of Dr. Robert Molday, University of British Columbia | | IF(1:500) |
| Antibody | anti-OPN1SW (Goat polyclonal) | Santa Cruz | Cat#: sc-14363 | IF(1:500) |
| Antibody | anti-GT335 (Mouse monoclonal) | AdipoGen | Cat#: AG-20B- 0020 | IF(1:500) |
| Antibody | anti-Rootletin (Chicken polyclonal) | PMID: 12427867 | | IF (1:200) |
| Antibody | anti-ARL13B (Rabbit polyclonal) | Proteintech | Cat#: 17711–1-AP | IF (1:500) |
| Antibody | anti-OPN1LMW (Rabbit polyclonal) | Millipore | Cat#: AB5405 | IF (1:500) |
| Antibody | anti-CEP290 (Rabbit monoclonal) | Bethyl laboratories | Cat#: A301-659A | WB (1:500) |
| Antibody | anti-GAPDH (Mouse monoclonal) | Millipore | Cat#: G8795 | WB (1:1000) |
| Antibody | anti-PDE6b (Rabbit polyclonal) | PMID: 33107904 | | IF (1:500) |
| Antibody | anti-Synaptophysin (Mouse monoclonal) | Abcam | ab8049 | IF (1:200) |
| Antibody | anti-p62 (Rabbit polyclonal) | Abcam | ab109012 | IF (1:200) WB (1:500) |
| Antibody | anti-LC3A/B (D3U4C) (Rabbit monoclonal) | Cell Signaling | 12741 | For human retinal organoids, WB (1:1000) |
| Antibody | anti-LC3B (Rabbit polyclonal) | Abcam | ab51520 | For mouse retina, WB (1:1000) |
| Antibody | anti-b-Actin (Mouse monoclonal) | Millipore | A5316 | WB (1:5000-1:10000) |
| Antibody | anti-20S (Rabbit polyclonal) | Enzo Life Sciences | BML-PW8155-0100 | WB (1:500) |
| Antibody | anti-IFT88 (Rabbit polyclonal) | Proteintech | 13967–1-AP | WB (1:500) |
| Antibody | anti-BBS6 (Rabbit polyclonal) | Michel Leroux | #17 | WB (1:500) |
| Antibody | anti-CEP164 (Rabbit polyclonal) | GeneTex | GTX85298 | WB (1:500) |

*Appendix 1 Continued on next page*

*Appendix 1 Continued*

| Reagent type (species) or resource | Designation | Source or reference | Identifiers | Additional information |
|---|---|---|---|---|
| Antibody | anti-OFD1 (Rabbit polyclonal) | Thermo | PA5-115684 | WB (1:500) |
| Antibody | anti-HDAC6 (Rabbit polyclonal) | Abgent | AP1106A | WB (1:1000) |
| Antibody | anti-RP1 (Chicken polyclonal) | PMID: 11773008 | | IF (1:2000) |
| Antibody | anti-Bassoon (Rabbit monoclonal) | Cell Signaling | 6897 S | IF (1:200) |
| Antibody | anti-CHX10 (Sheep polyclonal) | Abcam | ab16141 | IF (1:200) |
| Antibody | anti-PKCa (Rabbit monoclonal) | Thermo Fisher | MA1-157 | IF (1:1000) |
| Antibody | anti-CRALBP (Mouse monoclonal) | Abcam | ab15051 | IF (1:200) |
| Antibody | anti-GFAP (Rabbit polyclonal) | Dako | Z0334 | IF (1:500) WB (1:1000) |
| Antibody | anti-Calretinin (Rabbit monoclonal) | Millipore | MAB1568 | IF (1:200) |
| Commercial kits | Proteasome 20 S Activity Assay Kit | Sigma-Aldrich | MAK172-1KT | |
| Chemical compound, drug | Reserpine | Sigma Aldrich | 06859 | |
| Chemical compound, drug | MRT68921 | Sigma Aldrich | SML1644 | |
| Chemical compound, drug | Lys05 | Sigma Aldrich | SML2097 | |
| Chemical compound, drug | Chloroquine | Sigma Aldrich | C6628 | |
| Chemical compound, drug | Hydroxychloroquine | Sigma Aldrich | H0915 | |
| Chemical compound, drug | ROC-325 | Selleck | S8527 | |
| Chemical compound, drug | Bafilomycin A1 | Sigma Aldrich | 19–148 | |
| Chemical compound, drug | Tubastatin A | Sigma Aldrich | SML0044 | |
| Software, algorithm | Photoshop CC 2019 | Adobe | | |
| Software, algorithm | FIJI | FIJI | | |
| Software, algorithm | R | R-core team | | |
| Software, algorithm | Biowulf Linux cluster | National Institutes of Health | | http://biowulf.nih.gov |
| Other | DAPI stain | Invitrogen | D1306 | (1 µg/mL) |

