## [Editor Report]

This work provides an important pipeline for high-throughput screening platform to be used for drug discovery. The current data are complete. Validation of human patients-derived iPSC clones and functional assays in mice further strengthens the current conclusion.

---

## [Decision Letter]

**Decision letter after peer review:**

Thank you for submitting your article "Reserpine maintains photoreceptor survival in retinal ciliopathy by resolving proteostasis imbalance and ciliogenesis defects" for consideration by *eLife*. Your article has been reviewed by 3 peer reviewers, one of whom is a member of our Board of Reviewing Editors, and the evaluation has been overseen by a Reviewing Editor and Mone Zaidi as the Senior Editor. The following individual involved in review of your submission has agreed to reveal their identity: Marius Ueffing (Reviewer #2).

The current study proposed a drug discovery pipeline to accelerate the process of identifying drug candidates for LCA10 patients using mouse retinal organoid for initial screening, human patient iPSC-derived retinal organoid for further testing, and then mouse mutants for in vivo validation. Reserpine was identified as the top candidate, possibly through modulating proteostasis and autophagy to promote cilium assembly. The study is with high translational value. However, the rationale using dissociated single cells from mouse retinal organoid for drug screening needs to be justified. Further validation of human patient-derived iPSC clones and multiple clones to represent each patient are required. The consistency of phenotypic characteristics in human patient iPSC-derived retinal organoid needs to be reported. It was unclear if the rescued phenotypic changes were from the drug effects or a result of phenotypic variations in organoids. Functional assays in mice (ERG) could further improve the impact of the study: untreated control, treated-control, untreated mutant and treated-mutant in vivo.

Essential revisions:

1) Further validation of human patient-derived iPSC clones and multiple clones to represent each patient are required. The consistency of phenotypic characteristics in human patient iPSC-derived retinal organoid needs to be reported.

2) Functional assays in mice (ERG) could further improve the impact of the study: untreated control, treated-control, untreated mutant and treated-mutant in vivo.

3) The rationale using dissociated single cells from mouse retinal organoid for drug screening needs to be justified.

*Reviewer #1 (Recommendations for the authors):*

1. Figure 3: The number of human retinal organoids used for each experimental condition needed to be reported here. What are the differences in clinical symptoms between LCA-1 and LCA-2 patient?

2. Figure 6: The immunostaining signals could be affected by experimental procedure. In addition to the IHC results, functional tests of retinal activity would add more value into the validation of reserpine in rd16 mice.

3. Figure S5: Panel I was missing.

4. Figure S7: WT retinas shall be included to reveal inner retinal neuronal changes in rd16 mice.

5. In all figures, n number and statistics shall be included in the figure legends.

*Reviewer #2 (Recommendations for the authors):*

To further improve the manuscript, the following points could be addressed:

– Are there studies showing reserpine treatment in other retinal diseases? Please discuss.

– What about the use of DMSO as a vehicle control? Also, does DMSO cause certain side effects?

– According to Figure 3 and FS5, three time points were considered as reserpine treatments. Please explain additional reserpine treatments in the text. It is also not clear why the authors choose different days for the treatment for the RNA-seq experiment (from D117 to D150).

– Figure 3: Quantification of rhodopsin and opsin is required, as done in Figure 2.

– What about toxicity to monitor degeneration of photoreceptors in organoid and mouse models? Cell death assays may help to understand this. Otherwise, it would be difficult to say anything about photoreceptor cell death.

– Figure 6D- Quantification of p62 is missing. Also, please explain why WT organoids lack 20S protein expression.

– Figure S3- Analysis of fluorescence of RHO, synaptophysin, basson, and other proteins is needed to detect variability between samples.

– Did the authors check ubiquitin as a marker for UPS in organoids or mouse model before and after treatment?

– Why was 40 µM of reserpine chosen for intravitreal injection into mouse retina? Were other concentrations tested?

– "To experimentally examine the role of PN in survival of LCA10 photoreceptors, we supplemented organoid cultures with various autophagy inhibitors (MRT68921, Lys05, chloroquine, hydroxychloroquine, ROC-325) that target different steps of the autophagy pathway (Figure S6A)". Please indicate on which day and for how long the organoids were treated.

– There are three different bands in LC3 western blotting. Which band was thought to be LC3 II?

– How did the authors relate the reduced 20S expression to the increased proteasomal activity? Please explain.

– "Notably, reserpine treatment significantly reduced GFAP levels that expanded to the entire retina in untreated mice (Figure S7C),.." Please show quantification of GFAP.

– "Consistent with the moderate response of LCA-2 patient organoids to reserpine (Figure 3B), we did not identify significant differentially expressed (DE) genes between untreated and treated groups of LCA-2 for downstream analyses (data not shown) and thereby focused only on LCA-1.". Considering LCA-1 and LCA.2 are two affected compound heterozygous children, did authors see any differences between LCA-2 to LCA-1 via RNA-seq analysis?

– Some studies show that survival of autophagy is protective in retinal degeneration. Please discuss this in the Discussion.

– Discussion: The relationship between reserpine and p62 should be presented more clearly and should be consistent with previously published work.

– Recommendations for the authors:

As written in the public review, functional assays in mice (ERG) could significantly improve the impact of the study: untreated control, treated control, untreated mutant, and treated mutant in vivo.

– the quality of the images can be improved. Each figure should be labeled in the same order (e.g., red, green, blue).

– Not all antibodies are listed in the antibody list. Please complete.

–Figure 4 B: Please label the gene names and explain the heat map showing the up-regulated/down-regulated genes. The red rectangle highlights the "proteasome" factor. Highlight the rectangle there.

– Figure S5: The labels in the figure are not visible. Please update them.

Figure S6 B and D: Please analyze rhodopsin and S-opsin. Improve the quality of the images.

– Figure S7: Please use high quality images.

*Reviewer #3 (Recommendations for the authors):*

First, it is difficult to determine if the proper validations and controls were used for this paper since the methods section describing iPSC derivation refers to Shimada et al. 2017. And the methods description in that paper was not thorough.

The authors must provide proper karyotype characterization of all iPSC lines used in this study, including mouse lines. No conclusions can be made until the iPSC lines have been shown to have normal chromosomal structure. Chromosomal abnormalities are very common in iPSC derivation.

For the patient derived iPSCs it does not appear that the authors used multiple iPSC clones to represent each patient. This is standard practice for patient iPSC modeling and especially relevant in this manuscript as the authors did not detect differentially expressed genes in the LCA2 patient retinal organoids.

Since the LCA2 retinal organoids did not show gene expression differences even though they do show structural differences, so there is not enough data to conclude that the gene expression differences observed in LCA1 were due to the CEP290 mutation. Differences in gene expression could be due to differences in iPSC derivation. Authors need to show repeatable differences in multiple independently derived iPSC clones from the same patient, or multiple patients, or correct the mutation using CRISPR editing to show rescue.

The experiment as it stands is not controlled sufficiently enough to make conclusions.

---

## [Author Response]

Essential revisions:1) Further validation of human patient-derived iPSC clones and multiple clones to represent each patient are required. The consistency of phenotypic characteristics in human patient iPSC-derived retinal organoid needs to be reported.

We agree that validation of the patient induced pluripotent stem cell (iPSC) lines and the variability of clones is a concern for the drug effects on the derived retinal organoids. Therefore, we have now reported the karyotypes (Figure 1—figure supplement 1) and other quality controls (Materials and methods) in the manuscript and tested reserpine on retinal organoids derived from 2 clones of each (a total of 4) of LCA1 and LCA2 patients. As suggested by the reviewers, we quantified the fluorescence intensity of rod marker rhodopsin staining in multiple sections of at least two batches of differentiation in the treatment of reserpine and other relevant compounds to better present the variations. Reserpine treatment significantly increased the fluorescence intensity of rhodopsin in retinal organoids differentiated from multiple lines (Figure 3C). Although the increase of rhodopsin fluorescence intensity was not significant in some clones treated with reserpine, which could be due to their large variations, an obvious positive trend could still be observed in these organoids (Figure 3—figure supplement 2). We have also included immunoblotting analyses of key autophagy and primary cilium markers in untreated and treated LCA2 patient organoids (Figure 5—figure supplement 3). A consistent trend of changes of these markers between two patients was observed, indicating that the mechanisms of action of reserpine proposed in this manuscript could be applied for different *CEP290*-LCA patients regardless of the responsiveness to reserpine.

2) Functional assays in mice (ERG) could further improve the impact of the study: untreated control, treated-control, untreated mutant and treated-mutant in vivo.

We agree and have performed ERG experiments as suggested by the reviewer. The CEP290 disease in *rd16* mouse retina has a very early onset. Because of technical challenges in intravitreal injections in the small neonatal or early postnatal mouse eye, we performed two intravitreal injections at P7 and P14 and electroretinogram (ERG) data from untreated and treated wild type (WT) and *rd16* mice are now included in the manuscript (Figure 6—figure supplement 2 and Figure 6D). No significant differences were found in the morphology of rod and cone photoreceptors (Figure 6—figure supplement 2) or scotopic or photopic a-/b- waves (Figure 6—figure supplement 2) between WT retina injected with vehicle (DMSO) and reserpine, suggesting reserpine did not impact WT retina.

*Rd16* mice injected twice, at P7 and P14, with reserpine showed a marginally improved scotopic a-wave and a significantly higher b-wave, suggesting an improvement of rod photoreceptors in *rd16* mice by reserpine treatment (Figure 6D). No significant differences were detected in photopic a- or b- waves between untreated and treated *rd16* mice, likely because the cones are affected later.

Further development of therapies using reserpine and related molecules would require extensive optimization of dosage and delivery methods as well as toxicity studies. These are now subject of future investigations.

3) The rationale using dissociated single cells from mouse retinal organoid for drug screening needs to be justified.

Medium to high-throughput screening strategies require easy and consistent assay parameters at least during primary screens. We screened over 6000 small molecules at 11 different concentrations with duplicates.

Numerous studies from our and other labs have shown an early effect of the disease mutation on rod photoreceptors. We could visualize rods by GFP signal that was driven by the *Nrl* promoter. However, based on our flow cytometry analyses, *rd16* iPSC-derived retinal organoids show substantial variations in *Nrl*-GFP signal, which could mask positive hits in the primary screens. We therefore used single-cell cultures dissociated from *rd16* iPSC-derived retinal organoids for primary drug screens to ensure the presentation of each compound to homogenous cells.

Due to the large scale of screens, it is not feasible to perform this manually and the cells were dispensed into culture plates by a semi-automatic electronic dispenser. Mouse retinal organoids are too big to be dispensed and would be damaged during the process. Also, the screening was performed in 1536-well plates, and each well is too small to accommodate even a single mouse organoid. Therefore, we performed the screens using single cells.

We have clarified this in the text.

Reviewer #1 (Recommendations for the authors):1. Figure 3: The number of human retinal organoids used for each experimental condition needed to be reported here. What are the differences in clinical symptoms between LCA-1 and LCA-2 patient?

Each image was representative of at least 3 batches of differentiation, each of which had at least 2 organoids. This information was mentioned in Methods and Materials, but we have now included it in the figure legends as well. We also added a violin plot showing the fluorescent intensity of rhodopsin staining to better present the variations in each experiment (Figure 3C and Figure 3—figure supplement 2).

LCA-1 and LCA-2 patient displayed comparable clinical symptoms including almost undetectable full field electroretinogram, nystagmus (eye movements), poor pupillary light responses, and oculodigital reflex. Clinical phenotypes of *CEP290*-LCA patients have been described previously in many publications by Prof. Samuel G. Jacobson (U. Penn), who sadly passed away a few weeks ago. Only the retinal phenotypes will be represented in the organoids, as predicted, and likely show an early stage in disease pathogenesis.

Variability in drug response is common in human experiments because of extensive genetic variations. The response to reserpine in retinal organoids could be caused by variations in derivative iPSC lines, differentiation conditions, and/or intrinsic genetic variants between the two patients.

2. Figure 6: The immunostaining signals could be affected by experimental procedure. In addition to the IHC results, functional tests of retinal activity would add more value into the validation of reserpine in rd16 mice.

We thank the reviewers for the suggestions. The rescue of photoreceptors by reserpine on *rd16* mice was not just shown by immunostaining but primarily by the increase of outer nuclear layer thickness and the improvement of the outer segment biogenesis.

To strengthen our conclusions, we have now performed functional tests and shown that reserpine treatment increased scotopic a-wave marginally and b-wave significantly, suggesting an improvement of rod photoreceptor function (Figure 6D).

3. Figure S5: Panel I was missing.

We apologize. Due to the limited space in the Figure (as is generally the case), Panel I was placed next to Panel D instead of the end of the figure. This is now Figure 4—figure supplement 1 in the revised manuscript.

4. Figure S7: WT retinas shall be included to reveal inner retinal neuronal changes in rd16 mice.

Thanks for pointing this to us. Images from WT retina are included now (Figure 6—figure supplement 3).

5. In all figures, n number and statistics shall be included in the figure legends.

N number and statistics are now included in the figure legends.

Reviewer #2 (Recommendations for the authors):To further improve the manuscript, the following points could be addressed:– Are there studies showing reserpine treatment in other retinal diseases? Please discuss.

Reserpine has been reported to induce rod outer segment elongation. The systemic effects of reserpine on the modulation of outcome of intraocular pressure drug and allergic inflammation of the eye have also been reported. However, to our knowledge, there are no other studies testing reserpine in other retinal diseases. We have now briefly mentioned these in the Discussion.

– What about the use of DMSO as a vehicle control? Also, does DMSO cause certain side effects?

The percentage of DMSO used in the study is 0.4% (v/v) or less. In cell culture experiments, 0.5% DMSO as the final concentration has been used widely without cytotoxicity. DMSO is generally considered as non-toxic (in rat retina as high as 2%) and widely used as a vehicle control for intravitreal injections. We performed intravitreal injections of 0.4% DMSO in wild type and *rd16* mouse retina and did not observe any side effects by histologic evaluation or ERG (wild type data with DMSO and reserpine injections are shown in Figure 6—figure supplement 2). No systemic toxicity has been observed in the injected mice either.

– According to Figure 3 and FS5, three time points were considered as reserpine treatments. Please explain additional reserpine treatments in the text. It is also not clear why the authors choose different days for the treatment for the RNA-seq experiment (from D117 to D150).

We have now provided more details in the Methods and Material to explain the two reserpine treatments. To evaluate the effect of reserpine, the organoids were harvested 10 days after the third treatment to allow sufficient time for drug actions. The purpose of the RNA-seq experiments is to identify the signaling pathways modulated by reserpine. In this case, we harvested the organoids immediately after the third drug treatment to avoid compounding secondary effects.

– Figure 3: Quantification of rhodopsin and opsin is required, as done in Figure 2.

We thank the reviewers for suggestions. A violin plot with fluorescence intensity of rhodopsin is now provided at Figure 3C and Figure 3—figure supplement 2. As LCA patient organoids did not have dramatic phenotypes on cone photoreceptors, we did not quantify cone opsins and have modified the text accordingly.

– What about toxicity to monitor degeneration of photoreceptors in organoid and mouse models? Cell death assays may help to understand this. Otherwise, it would be difficult to say anything about photoreceptor cell death.

Here, we have focused primarily on rod photoreceptor survival as an assay and started treatment before cell death occurs (i.e., day 107 in organoids and P7 in mouse retina). For human retinal organoids, no dramatic difference of outer nuclear layer thickness on patient organoids have been observed by us or reported by other groups at this stage. Cell death in *rd16* mouse retina is detected at or after P12. Therefore, the increase of outer nuclear layer observed in reserpine-treated retina suggests an inhibition of photoreceptor cell death as post-mitotic rod photoreceptors do not proliferate at this stage.

We will be performing extensive dosage and drug toxicity experiments as part of our future investigations.

– Figure 6D- Quantification of p62 is missing. Also, please explain why WT organoids lack 20S protein expression.

We have now added quantification of p62 puncta (Figure 7B). The 20S protein expression was from WT mice but not organoids. 20S protein is present in WT retinal lysate but detected in a much lower amount compared to the *rd16* ones. Overexposure of the immunoblotting membrane could reveal the presence of 20S expression. In concordance, we had previous shown the expression of 20S proteosome in RNA-seq data of rod photoreceptors (Kim et al., *Cell Reports*, 2016).

– Figure S3- Analysis of fluorescence of RHO, synaptophysin, basson, and other proteins is needed to detect variability between samples.

We have now included the quantification of rhodopsin staining in reserpine and other compound treatments as reserpine mainly acts on rod photoreceptors. We did not quantify the immunostaining of other retinal cell types as the high variation of them make it technically challenging and labor intensive to reach a solid conclusion, yet we note that it is not the main goal of the manuscript. We performed immunostaining of other retinal or structural markers only to demonstrate that reserpine did not largely impact other retinal structures or cell types. We have modified the text and removed the descriptive comparison of other retinal cell types or structure between untreated and treated organoids.

– Did the authors check ubiquitin as a marker for UPS in organoids or mouse model before and after treatment?

We did not compare ubiquitin level as ubiquitin-labeled proteins could also be targeted by autophagy and are not specific to UPS.

– Why was 40 µM of reserpine chosen for intravitreal injection into mouse retina? Were other concentrations tested?

In the primary screens, the EC_50_ of reserpine was around 20 uM. We doubled the concentration for injection to account for the potential loss of reserpine during the procedures. As we observed a rescue effect of reserpine, we used the same concentration for all experiments at this stage as animal work takes a long time. No other concentrations of reserpine were tested in the current study.

We will be performing extensive dosage and drug toxicity experiments as part of our future investigations.

– "To experimentally examine the role of PN in survival of LCA10 photoreceptors, we supplemented organoid cultures with various autophagy inhibitors (MRT68921, Lys05, chloroquine, hydroxychloroquine, ROC-325) that target different steps of the autophagy pathway (Figure S6A)". Please indicate on which day and for how long the organoids were treated.

Treatment of other autophagy inhibitors was performed the same way as reserpine treatment. We have now provided this information in the figure and text.

– There are three different bands in LC3 western blotting. Which band was thought to be LC3 II?

We apologize for the confusion. The lower band is LC3-II. We have added labels in our immunoblotting images.

– How did the authors relate the reduced 20S expression to the increased proteasomal activity? Please explain.

The expression level of 20S was consistently elevated in patient organoids and *rd16* mouse retina, yet the chymotrypsin-like activities were lower compared to the control, suggesting the higher expression of 20S protein level was to compensate for their low activity. While reserpine treatments partially restored proteostasis and increased 20S activity, the high level of 20S was not needed and thus was reduced.

We have added this in the Discussion.

– "Notably, reserpine treatment significantly reduced GFAP levels that expanded to the entire retina in untreated mice (Figure S7C),.." Please show quantification of GFAP.

The expression level of GFAP is now shown by immunoblotting analyses and quantification (Figure 6—figure supplement 3).

– "Consistent with the moderate response of LCA-2 patient organoids to reserpine (Figure 3B), we did not identify significant differentially expressed (DE) genes between untreated and treated groups of LCA-2 for downstream analyses (data not shown) and thereby focused only on LCA-1.". Considering LCA-1 and LCA.2 are two affected compound heterozygous children, did authors see any differences between LCA-2 to LCA-1 via RNA-seq analysis?

We did observe differences in RNA-seq data between LCA-1 and LCA-2. Although the differentially expressed genes compared to the control were largely similar between the two patients, the expression level and the peak of appearance of these DE genes was different. This could also be one explanation as to why reserpine only showed moderate effect on LCA-2 organoids.

– Some studies show that survival of autophagy is protective in retinal degeneration. Please discuss this in the Discussion.

The effect of autophagy on retinal degeneration depends on the cause of the disease. Balance of proteostasis pathways is critical for cell survival. In addition, we argue that, regardless of autophagy activation or inhibition, it was the clearance of accumulative proteins that rescues photoreceptors. We have modified the Discussion to emphasize this point.

– Discussion: The relationship between reserpine and p62 should be presented more clearly and should be consistent with previously published work.

To our knowledge, we had a consistent outcome of p62 accumulation as previous studies. We have modified the Discussion to discuss their relationship and subsequent impact on proteostasis.

– Recommendations for the authors:As written in the public review, functional assays in mice (ERG) could significantly improve the impact of the study: untreated control, treated control, untreated mutant, and treated mutant in vivo.

We thank the reviewer for the suggestions. As mentioned previously, we have now performed ERG experiments to compare untreated/treated WT/*rd16* mice. No significant differences were found between untreated and treated WT mice, suggesting the reserpine treatment did not interfere retinal functions of WT mice (Figure 6—figure supplement 2). Functional improvement of rod photoreceptors in *rd16* mice was revealed by marginally increased a-wave and significantly increased b-wave (Figure 6D).

– the quality of the images can be improved. Each figure should be labeled in the same order (e.g., red, green, blue).

We used high-quality images for all our figures. The obscure images in the supplemental figures could be due to the process of generation of the PDF file. We have improved our way to present the figures. We also modified the order of each channel across all images for improved consistency.

– Not all antibodies are listed in the antibody list. Please complete.

We apologize and have now completed the antibody list (Key Resources Table).

–Figure 4 B: Please label the gene names and explain the heat map showing the up-regulated/down-regulated genes. The red rectangle highlights the "proteasome" factor. Highlight the rectangle there.

The heatmap in Figure 4B summarizes the expression of 355 genes responding to reserpine treatment and thus names cannot be added due to the large number of genes. We have now included Table Supplement 1 with details of the differentially expressing genes. Additionally, Figure 4C now has names of the top differentially expressed genes labeled in the volcano plot. We also added a red rectangle highlighting Proteasome.

– Figure S5: The labels in the figure are not visible. Please update them.

We thank the reviewer for pointing this out. We have adjusted the size of the figure to improve legibility.

Figure S6 B and D: Please analyze rhodopsin and S-opsin. Improve the quality of the images.

We have now quantified the fluorescence intensity of rhodopsin in the images and improved the export of images. S-opsin staining did not show morphological changes and is not the focus of the manuscript. We have modified the text accordingly.

– Figure S7: Please use high quality images.

Our previous way of generation of the PDF file for supplemental figures may have blurred the images. We have now improved our way to generate the PDF.

Reviewer #3 (Recommendations for the authors):First, it is difficult to determine if the proper validations and controls were used for this paper since the methods section describing iPSC derivation refers to Shimada et al. 2017. And the methods description in that paper was not thorough.The authors must provide proper karyotype characterization of all iPSC lines used in this study, including mouse lines. No conclusions can be made until the iPSC lines have been shown to have normal chromosomal structure. Chromosomal abnormalities are very common in iPSC derivation.For the patient derived iPSCs it does not appear that the authors used multiple iPSC clones to represent each patient. This is standard practice for patient iPSC modeling and especially relevant in this manuscript as the authors did not detect differentially expressed genes in the LCA2 patient retinal organoids.Since the LCA2 retinal organoids did not show gene expression differences even though they do show structural differences, so there is not enough data to conclude that the gene expression differences observed in LCA1 were due to the CEP290 mutation. Differences in gene expression could be due to differences in iPSC derivation. Authors need to show repeatable differences in multiple independently derived iPSC clones from the same patient, or multiple patients, or correct the mutation using CRISPR editing to show rescue.The experiment as it stands is not controlled sufficiently enough to make conclusions.

We have now provided more details on iPSC derivation, maintenance, quality control, and differentiation. Karyotypes of all mouse and human iPSC lines were provided in Figure 1—figure supplement 1. The karyotyping analysis was independently performed by experienced personnel in the Comparative Cytogenetics Core Facility of National Cancer Institute. Cells were harvested according to the standard protocol and prepared for G-band stain. Fifteen unique metaphases were analyzed. Retinal organoids were generated using iPSC lines within 10 passages of test cells. As shown in Figure 1—figure supplement 1, all mouse and human iPSC lines exhibited normal karyotypes except *rd16* line 2, which had one missing sex chromosome. As one missing sex chromosome should not have dramatic impact on disease-associated phenotypes, we kept *rd16* line 2 to confirm the phenotypes of *rd16* organoids but used *rd16* line 1, which was karyotypically normal, for high-throughput screening and secondary assays.

We have now tested reserpine on retinal organoids derived from 2 clones of each of LCA1 and LCA2. We also quantified the fluorescence intensity of rod marker rhodopsin staining in multiple sections of at least two batches of differentiation (Figure 3C and Figure 3—figure supplement 2). A marginally or significantly increase of rhodopsin fluorescence intensity could be observed in retinal organoids differentiated from multiple clones and cell lines.

In our manuscript, we used retinal organoids differentiated from *CEP290*-LCA as a model to identify drugs to maintain photoreceptor survival. We hypothesized that reserpine treatment is not directly rescuing CEP290 function but works downstream to keep the cells alive, and we performed RNA-seq to identify the pathways that might be implicated in this process. Our data provided the clues that we pursued to understand the mechanism. We performed additional assays to validate the dysregulated proteostasis pathway identified in RNA-seq data. Further validation of other pathways in different models are needed but beyond the scope of this manuscript. Therefore, we do not believe that additional RNA-seq on multiple cell lines or corrected CEP290 clones will provide further insights into how reserpine treatment leads to rod photoreceptor survival and maintenance of function.

As reserpine does not act as a transcription factor, the change of gene profiles upon reserpine treatment could vary in time and intensity, which could explain the few differentially expressed genes observed in LCA-2. The differentially expressed genes being shown here are between untreated and treated patient organoids but not between control and patients. Notably, although LCA2 did not show sufficient differentially expressed genes, dysregulation of proteostasis pathway is still observed in LCA patients and the action mechanism of reserpine could be validated in LCA2 (Figure 5—figure supplement 3), further strengthening our findings.

Our manuscript has focused on identification of drug candidates for *CEP290*-LCA disease. As mentioned previously, “We would also like to respectfully point out that after the initial medium-high throughput small molecule drug screening using *rd16* rod photoreceptors, we performed secondary screening of selected compounds using *rd16* mouse retinal organoids. Then, we performed reserpine (lead compound) treatment studies using human retinal organoids from CEP290-LCA patients and in vivo in *rd16* mouse retina. We have now added additional experiments using human patient organoids as well as functional data from *rd16* mouse. Thus, reserpine has gone through 4 different screens and validations. The functional recovery of rod photoreceptors in reserpine-treated *rd16* mice and comparable action mechanisms of reserpine on in vivo *rd16* mice demonstrates the success of our drug discovery pipeline and the validations of our findings.